# The unfolded protein response and endoplasmic reticulum protein targeting machineries converge on the stress sensor IRE1

Diego Acosta-Alvear[1,2†‡*], G Elif Karagöz[1,2†§*], Florian Fröhlich[3,4#], Han Li[1,2], Tobias C Walther[1,3,4], Peter Walter[1,2*]

[1]Howard Hughes Medical Institute, United States; [2]Department of Biochemistry and Biophysics, University of California, San Francisco, San Francisco, United States; [3]Harvard School of Public Health, Harvard Medical School, Boston, United States; [4]Department of Cell Biology, Harvard Medical School, Boston, United States

*For correspondence:
daa@lifesci.ucsb.edu (DAA);
elif.karagoez@univie.ac.at (GEK);
peter@walterlab.ucsf.edu (PW)

†These authors contributed equally to this work

Present address: ‡Department of Molecular, Cellular and Developmental Biology, University of California, Santa Barbara, Santa Barbara, United States; §Max F. Perutz Laboratories, Medical University of Vienna, Vienna, Austria; #Department of Biology/Chemistry, Molecular Membrane Biology Group, University of Osnabrück, Osnabrück, Germany

Competing interests: The authors declare that no competing interests exist.

**Abstract** The protein folding capacity of the endoplasmic reticulum (ER) is tightly regulated by a network of signaling pathways, known as the unfolded protein response (UPR). UPR sensors monitor the ER folding status to adjust ER folding capacity according to need. To understand how the UPR sensor IRE1 maintains ER homeostasis, we identified zero-length crosslinks of RNA to IRE1 with single nucleotide precision in vivo. We found that IRE1 specifically crosslinks to a subset of ER-targeted mRNAs, SRP RNA, ribosomal and transfer RNAs. Crosslink sites cluster in a discrete region of the ribosome surface spanning from the A-site to the polypeptide exit tunnel. Moreover, IRE1 binds to purified 80S ribosomes with high affinity, indicating association with ER-bound ribosomes. Our results suggest that the ER protein translocation and targeting machineries work together with the UPR to tune the ER's protein folding load.
DOI: https://doi.org/10.7554/eLife.43036.001

## Introduction

Protein folding and maturation in the endoplasmic reticulum (ER) are essential for cell physiology, as most of the all secreted and transmembrane proteins are synthesized and folded in this organelle. Perturbations leading to protein folding defects in the ER –collectively known as ER stress– activate an ensemble of transcriptional programs known as the unfolded protein response (UPR) (*Karagoz and Acosta-Alvear D, 2018*; *Walter and Ron, 2011*). The UPR maintains the health of the secreted and membrane-embedded proteome through (i) decreasing ER client protein load, (ii) upregulating chaperones and enzymes that assist protein folding, and (iii) promoting the degradation of misfolded proteins. Three ER membrane embedded protein folding sensors control the UPR: ATF6 (activating transcription factor 6), PERK (protein kinase R (PKR)-like kinase) and IRE1 (inositol requiring enzyme 1). IRE1 is the most evolutionarily conserved sensor and is found from yeast to metazoans. Unfolded proteins serve as direct ligands for IRE1's lumenal sensor domain, promoting its oligomerization and activation in the plane of the ER membrane (*Aragón et al., 2009*; *Gardner and Walter, 2011*; *Karagöz et al., 2017*; *Kimata et al., 2007*; *Li et al., 2010*). Active IRE1 molecules *trans*-autophosphorylate, allowing the subsequent allosteric activation of its C-terminal endoribonuclease domain (*Karagöz et al., 2017*; *Korennykh et al., 2011*). Active IRE1 responds to ER stress in two ways: (i) it cleaves an unconventional intron from the mRNA encoding the transcription factor XBP1 (X-box binding protein 1), initiating a spliceosome-independent mRNA splicing reaction that culminates in the production of XBP1s ('s' for spliced), a potent transcription activator

that increases the ER folding and degradation capacities (*Acosta-Alvear et al., 2007*; *Reimold et al., 2000*; *Reimold et al., 2001*; *Yoshida et al., 1998*), and (ii) it cleaves ER-targeted mRNAs in a process known as RIDD (regulated IRE1-dependent decay), thus lowering the ER protein-folding load (*Hollien et al., 2009*; *Hollien and Weissman, 2006*). In both cases, IRE1 substrate RNAs must be properly targeted to the ER membrane to meet the IRE1 enzyme.

In metazoans, the *XBP1* mRNA is brought to the ER membrane as part of the ribosome-nascent chain complex actively translating XBP1u ('u' for unspliced) (*Yanagitani et al., 2009*; *Yanagitani et al., 2011*). Peptide sequences encoded in the *XBP1u* mRNA are recognized by the signal recognition particle (SRP)-dependent co-translational targeting machinery, which is in charge of sorting of most of the mRNAs encoding ER resident, secretory and transmembrane proteins (*Plumb et al., 2015*). The engagement of the translating XBP1u protein with SRP results in its proper sorting to the ER membrane where it meets IRE1. Despite the progress made in unraveling the ER targeting mechanism of the *XBP1* mRNA, our current understanding of recruitment of IRE1 to membrane-bound *XBP1* mRNA or RIDD substrate mRNAs is still a mystery. Moreover, the precise determinants controlling IRE1's exquisite specificity remain limited to our understanding of *XBP1* mRNA and a small number of canonical RIDD substrates. Since the physiological consequences of IRE1 activation are the product of cleavage of ER-targeted transcripts, there is an inherent need to address the issue of substrate selection. Previous work aimed at identifying IRE1 substrate recognition relied on transcriptomics (*Han et al., 2009*; *Hollien et al., 2009*; *Hollien and Weissman, 2006*; *So et al., 2012*). These methods are limited as they cannot identify physical associations between IRE1 and its putative substrates, nor they can dissociate direct and indirect effects emanating from IRE1's nuclease activity.

To gain insights into these questions, we used complementary system levels and biochemical methods to identify both the RNAs and proteins that physically associate with IRE1 in living cells. Our approaches revealed novel interactions between IRE1 and select ER-bound RNAs, as well as between IRE1 and the co-translational protein targeting and translocation machineries. Taken together, our results link the UPR and the ER protein targeting machineries, establishing that these processes unexpected converge on IRE1.

## Results

### IRE1 associates with select RNAs

To identify RNAs that directly associate with IRE1 in living cells, we used photoactivatable-ribonucleoside-crosslinking and immunoprecipitation (PAR-CLIP) (*Hafner et al., 2010*). PAR-CLIP employs incorporation of photoactivatable ribonucleosides (*e.g.*, 4-thiouridine) into RNAs *in vivo*. Long-wavelength UV irradiation (365 nm) generates 'zero-length' crosslinks between the RNAs and proteins. In this way we identified RNAs crosslinked to IRE1 by deep sequencing (*Figure 1A*, *Figure 1—figure supplement 1A*). Towards this goal, we expressed an epitope-tagged (triple-FLAG-hexa-histidine) version of IRE1 ectopically from a tetracycline-inducible promoter in a human embryonic kidney derived cell line (HEK293Trex) and compared conditions with and without additionally imposed ER stress, induced by tunicamycin, a classical ER protein-folding poison that blocks *N*-linked glycosylation in the ER lumen. To optimize the signal to noise ratio, we fractionated cell lysates to enrich for membrane proteins (*Figure 1A*, *Figure 1—figure supplement 1A*, *Figure 1—figure supplement 1B*), and carried out the immunoprecipitations (IP) after fully denaturing the fraction by heating in SDS (*Figure 1—figure supplement 1A*, *Figure 1—figure supplement 1C*). We then subjected the immune-complexes to repeated washes in high-salt buffers.

Addition of doxycycline (a derivative of tetracycline) forced overexpression of IRE1 in the absence of ER stress inducers and resulted in its activation as evidenced by *XBP1* mRNA splicing (*Figure 1—figure supplement 1D*). IRE1 activity was further boosted by tunicamycin, as demonstrated by a pronounced increase in *XBP1* mRNA splicing (*Figure 1—figure supplement 1D*). We next ascertained that we recovered IRE1-RNA complexes under denaturing conditions in the presence of RNase to fragment associated RNAs into small oligonucleotides. To this end, we compared the efficiency of pull-down under denaturing and native conditions (*Figure 1—figure supplement 1C*). We confirmed that RNA was recovered by treating the immunoprecipitates with phosphatase and radiolabelling with polynucleotide kinase to mark the 5'-ends of IRE1-associated RNA fragments (*Figure 1—figure*

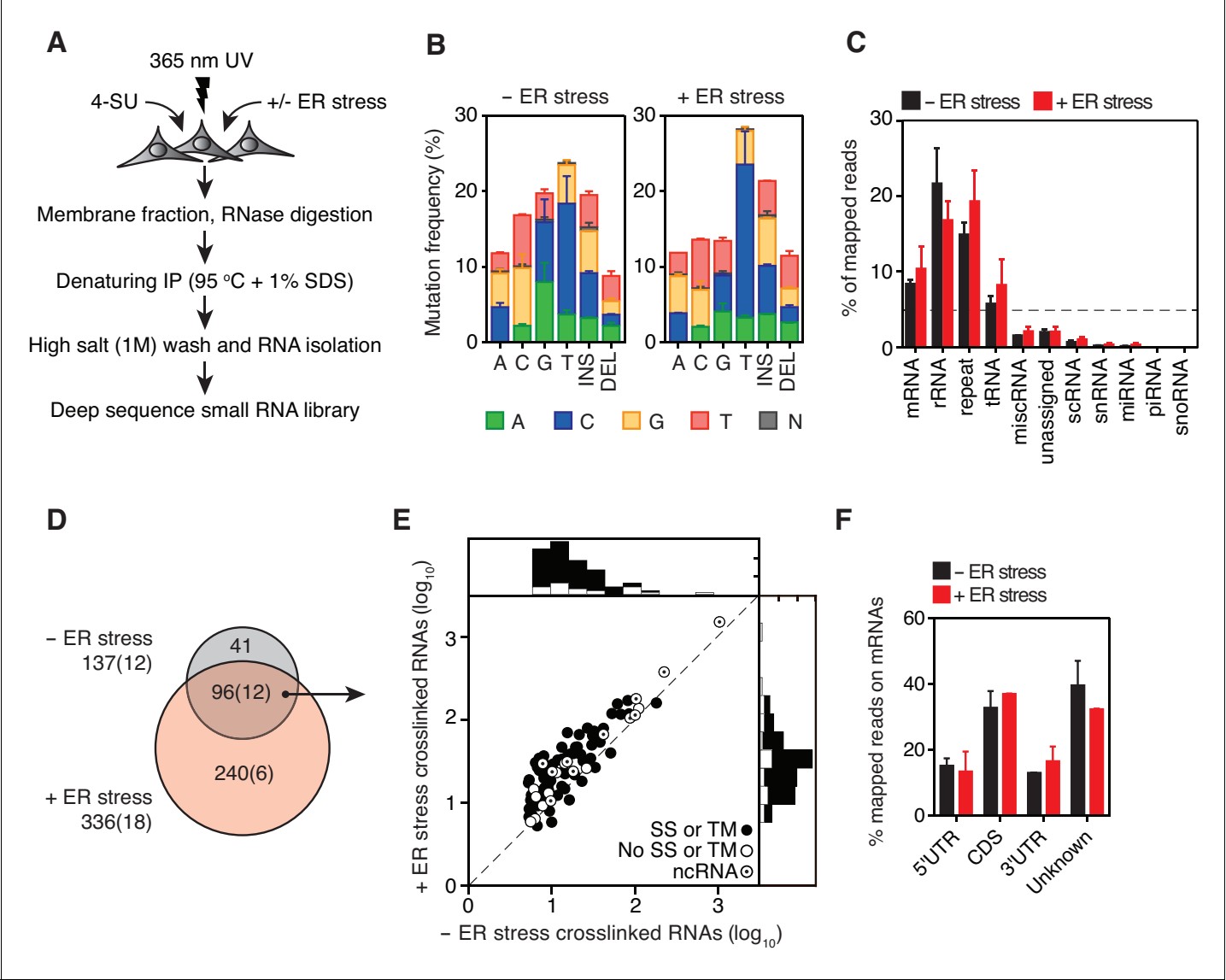

**Figure 1.** Identification of IRE1-associated RNAs. (**A**) Experimental strategy. (**B**) Mutation plots showing T→C transitions are the most common mutations recovered by PAR-CLIP in biological duplicates. (**C**) Breakdown of RNA classes associated with IRE1 identified by PAR-CLIP in biological duplicates. (**D**) Venn diagrams showing the numbers of crosslinked transcripts recovered by PAR-CLIP in the presence or absence of chemically induced ER stress in biological duplicates. Cardinals indicate the total or non-coding (in parenthesis) number of transcripts in each group. (**E**) Scatter plot showing the abundance of PAR-CLIP recovered transcripts in the presence or absence of chemically induced ER stress (copies per million reads: geometric mean of copy number per transcript in biological duplicates). The histograms above and to the side of the scatter plot illustrate the frequency of transcripts that traverse the secretory pathway. The diagonal dashed line indicates a slope of 1. (**F**) Breakdown of mRNA regions associated with IRE1 in PAR-CLIP experiments. 'Unknown region' refers to transcripts associated with coding loci but that do not have a single annotated coding sequence (*i.e.*, alternative splicing or alternative transcription initiation sites).

DOI: https://doi.org/10.7554/eLife.43036.002

The following figure supplements are available for figure 1:

**Figure supplement 1.** A modifed PAR-CLIP protocol for identifying IRE1-associated RNAs.
DOI: https://doi.org/10.7554/eLife.43036.003

**Figure supplement 2.** Reproducible identification of IRE1-crosslinked RNAs using our modified PAR-CLIP protocol.
DOI: https://doi.org/10.7554/eLife.43036.004

*supplement 1E*). Separation of the radiolabeled immunoprecipitates by SDS-PAGE verified the presence of crosslinked IRE1-RNA species migrating at the expected molecular weight of IRE1 (*Figure 1—figure supplement 1F*).

In the PAR-CLIP procedure, the generation of RNA-protein adducts results in the introduction of specific mutations at the crosslink sites, which can be identified most often as T→C transitions by deep sequencing upon mapping to a reference genome (*Hafner et al., 2010*) (*Figure 1—figure supplement 1A*). In our experiments, T→C transitions were the most prevalent mutations observed in biological duplicates (14.7 ± 3.6%, and 20.2 ± 4.4% of all mapped mutations in the absence and presence of ER stress, respectively, *Figure 1B* blue bars in columns labeled 'T'), strongly indicating recovery of IRE1-crosslinked RNA tags. We henceforth focused exclusively on analysis of the clusters that contained reads with T→C mutations.

In our analyses we first mapped the sequence reads derived from crosslinked IRE1-RNA complexes to the human genome to identify clusters over the length of each annotated transcript. We filtered the data based on a set of rules setting an arbitrary cut-off defined by a minimal number of reads bearing T→C transitions that must be contained in each cluster. We then summed the number of reads per cluster that fulfilled these criteria to estimate the relative abundance of IRE1-crosslinked reads per transcript, providing us with a ranked list of IRE1 crosslinked RNAs (see Materials and methods and *Figure 1—figure supplement 2A*). In these analyses, we identified a set of 137 IRE1-associated RNAs in the absence and 336 in the presence of ER stress, indicating that IRE1-RNA association increases with ER stress (*Figure 1C–E* and *Supplementary file 1*).

As expected, classification of the mapped reads revealed that IRE1 associated with a select set of mRNAs. Most of these mRNAs (86% of the mRNAs identified in the absence and 74% of the mRNAs identified in the presence of ER stress) encode proteins that traverse the secretory pathway (labeled in *Figures 1E* and *2A* by black circles, and *Supplementary file 1*). The most prominent interactions of RNAs with IRE1 occurred within a core-set of 96 RNAs found under both conditions, that is, with and without chemically induced ER stress (*Figures 1D* and *2A*, and *Supplementary file 1*).

Interestingly, we found that IRE1 not only crosslinked to mRNAs as expected but also to non-coding RNAs (12 of the 137 IRE1-crosslinked RNAs in the absence and 18 of the 336 IRE1-crosslinked RNAs in the presence of ER stress). These RNAs included long non-coding RNAs (*e.g.*, *MALAT1*, *XIST*), microRNA precursors, and small cytosolic RNAs (*e.g.*, Y-RNAs, vault RNAs, and the SRP RNAs) (*Figures 1C–1E* and *2A* and *Supplementary file 1*). Together, these data suggest that IRE1 not only interacts with mRNAs but that a larger spectrum of RNAs also associate with IRE1 in response to increasing amounts of ER stress.

We reproducibly identified a well-correlated group and ranking of transcripts in biological replicates (*Figure 1—figure supplement 2B*). Notably, our analyses revealed no correlation between transcript length and the abundance of crosslinked RNAs recovered (*Figure 1—figure supplement 2C*), indicating that IRE1-RNA contacts occur at a few discrete sites. As expected, we found no correlation between the fraction of reads bearing T→C transitions and the level of ER stress (*Figure 1—figure supplement 2D*). Therefore, the increase/decrease in read abundance rather than the change in frequency of T→C mutations are the best indicators of IRE1-RNA associations. Single transcript analysis validated that T→C transitions highly selectively converge on discrete sites (*Figure 2—figure supplement 1*).

We initially excluded rRNAs and tRNAs at this stage from the analysis because their high abundance skewed the results. However, guided by our discovery of IRE1 association with non-coding RNAs we decided to revisit the analysis of rRNAs and tRNAs independently as we describe below.

## IRE1 association and mRNA abundance do not correlate

Notably, IRE1 showed an apparent preference for coding over untranslated regions in the position of the mRNA crosslinks (*Figure 1F*), suggesting that the mRNA-IRE1 associations identified here occur preferentially as these mRNAs are being translated. Gene ontology (GO) analyses on the crosslinked mRNAs revealed a strong preference for transcripts encoding transmembrane proteins (*Figure 2B*), further underscoring the selectivity of the crosslinking reaction.

The association of IRE1 with mRNAs encoding proteins that traverse the secretory pathway (*Supplementary file 1*) suggested that crosslinking reflects IRE1 engagement with RIDD substrate mRNAs. Therefore, we next asked whether the select mRNAs that we identified by PAR-CLIP are degraded in an IRE1-dependent manner. Towards this end, we measured RNA levels by RNA-seq

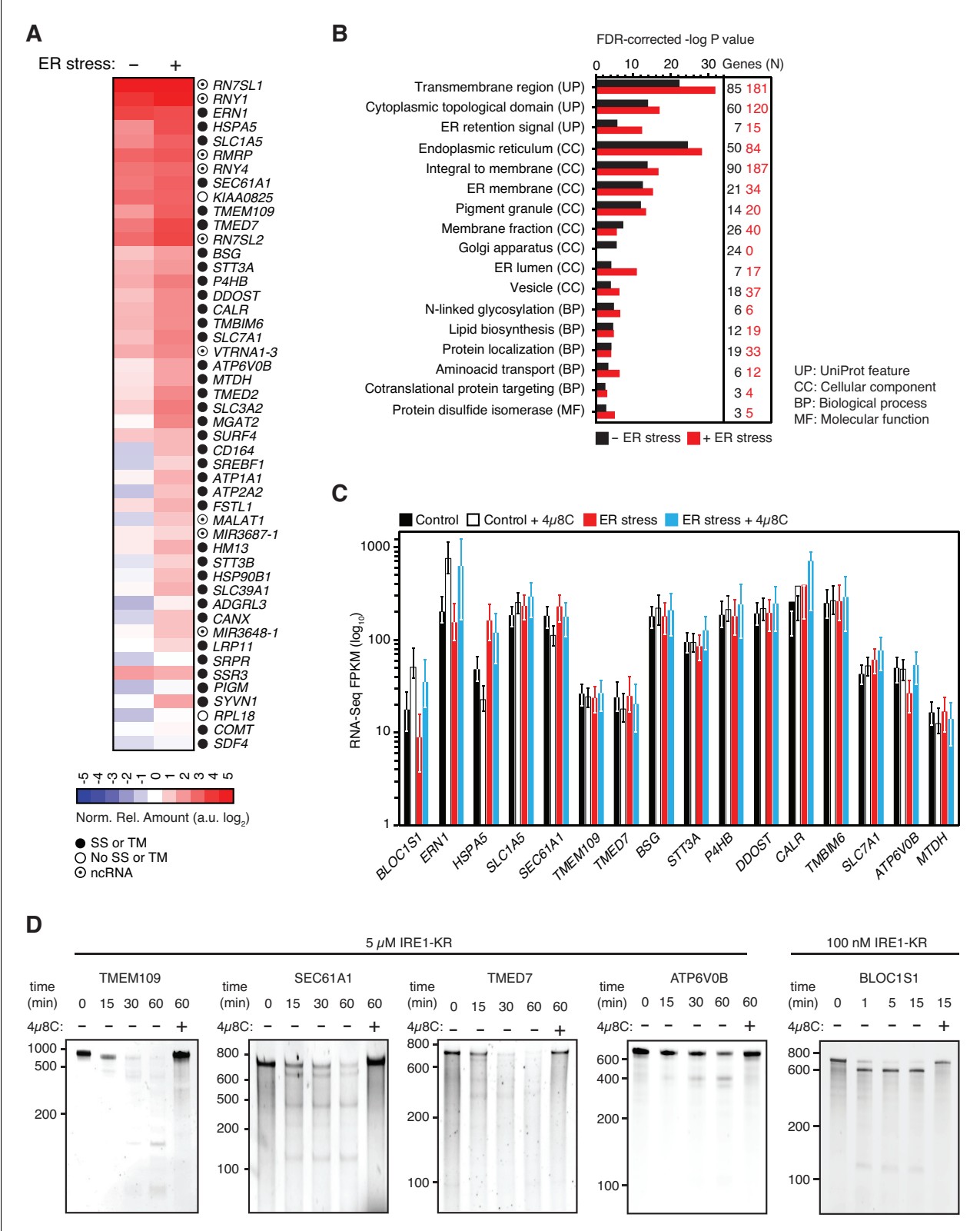

**Figure 2.** IRE1 associates with mRNAs and ncRNAs. (**A**) Heat map showing the relative amounts of select transcripts in the common-core (intersection of Venn diagrams. Arbitrary threshold: copies ≥ 10). (**B**) Gene-functional categories enriched among the crosslinked protein-coding transcripts recovered in each experimental set-up grouped by gene-ontology (GO) terms. (**C**) Changes in the transcript levels of select PAR-CLIP targets during ER

*Figure 2 continued on next page*

Figure 2 continued

stress measured by RNA-seq. Data: Mean FPKM values of biological duplicates. Error bars: 95% CI. (**D**) TBE-urea PAGE gels showing the IRE1-dependent cleavage of select PAR-CLIP targets. The IRE1 specific inhibitor 4μ8C (10 μM) was used as a control for specificity.

DOI: https://doi.org/10.7554/eLife.43036.005

The following figure supplements are available for figure 2:

**Figure supplement 1.** Examples of coverage tracks of PAR-CLIP sequencing reads mapped to coding loci in the presence or absence of chemically induced ER stress.

DOI: https://doi.org/10.7554/eLife.43036.006

**Figure supplement 2.** Expression profiles of IRE1-associated RNAs identified by PAR-CLIP.

DOI: https://doi.org/10.7554/eLife.43036.007

under the same ER stress conditions and time frames that we used for the PAR-CLIP experiments. We found a positive correlation between the expression level of each transcript and the transcript abundance recovered by PAR-CLIP. Importantly however, we neither observed a preferential association between IRE1 and the most highly expressed transcripts (*i.e.*, those at the tail end of the distribution, (*Figure 2—figure supplement 2A*), nor did we find a preferential association between IRE1 and the RNAs whose expression levels changed in response to ER stress (*Figure 2—figure supplement 2B and C*). Even though IRE1 association and mRNA abundance did not strongly correlate, we observed significant associations between IRE1 and a pool of abundant RNAs that include rRNAs, tRNAs, and SRP RNA, among others.

Moreover, the RNA-Seq results indicated that the levels of the majority of the IRE1-associated mRNAs did not change significantly in response to chemically induced ER stress in the time frames examined here (*Figure 2C* and *Figure 2—figure supplement 2C*). Taken together, these data suggest that either IRE1 engagement as detected by crosslinking does not lead to their cleavage by RIDD or that the fraction of cleaved mRNAs is too small to be detected by our methods.

To rule out that the observed effects on RNA levels stem from transcriptional changes, we treated different cell lines with the RNA polymerase inhibitor actinomycin D to prevent the accumulation of newly synthesized mRNAs and subjected these cells to chemical ER stressors in the presence or absence of the IRE1 inhibitor 4μ8C (*Cross et al., 2012*). In agreement with the RNA-Seq data shown above, we did not find evidence of significant IRE1-dependent degradation of select transcripts identified by PAR-CLIP. By contrast, we detected degradation of the canonical RIDD substrate *BLOC1S1* mRNA (*Figure 2—figure supplement 2D*). Thus, in the time window coincident with ER stress induction, as monitored by *XBP1* mRNA splicing and *BLOC1S1* mRNA degradation, the levels of the mRNAs captured by PAR-CLIP remained largely unaltered.

By contrast to this general trend, a few mRNAs exhibited modest IRE1-dependent decreases in their levels that were reversed by treatment with 4μ8C (*e.g.*, *ERN1* and *ATP6V0B* in *Figure 2C* and most mRNAs in *Figure 2—figure supplement 2E*; compare blue to red bars), suggesting that the crosslinked transcripts represent a population of suboptimal IRE1 RIDD substrates with longer RNA-protein dwell times that facilitate their capture by photo-crosslinking. To test this possibility, we performed *in vitro* RNA cleavage assays for a subset of highly enriched PAR-CLIP-captured mRNAs that included *TMEM109, SEC61A1, TMED7*, and *ATP6V0B* (*Figure 2A*). In these experiments, we used *in vitro* transcribed RNAs corresponding to either full-length mRNAs or artificial transcripts of similar lengths that contain the IRE1 crosslink sites identified by PAR-CLIP (*Figure 2—figure supplement 1*). We incubated these RNAs with recombinant IRE1 cytosolic kinase-RNase domains, heretofore referred as 'IRE1-KR' (*Figure 2D*). Our analyses showed that, by contrast to the *XBP1* and *BLOC1S1* mRNAs, which are processed by IRE1 with high efficiency, the PAR-CLIP captured mRNAs are cleaved by IRE1-KR at least one order-of-magnitude less efficiently, requiring longer incubation times and higher enzyme concentrations (*Figure 2D* and *Peschek et al., 2015*). These results suggest that the intrinsic properties of the RNAs determine the extent of cleavage by IRE1 in cells.

## IRE1 interacts with SRP RNA, select tRNAs and ribosomal RNAs

Among the transcripts found in the core-set of IRE1-interacting RNAs, we identified *RN7SL*, the RNA component of the SRP, as the most robust PAR-CLIP hit (*Figure 2A*). Topographical mapping of the crosslinking sites revealed that IRE1 preferentially interacted with the Alu domain of the SRP-RNA (*Figure 3*). Importantly, we also recovered reads mapping to the central S domain of the SRP RNA

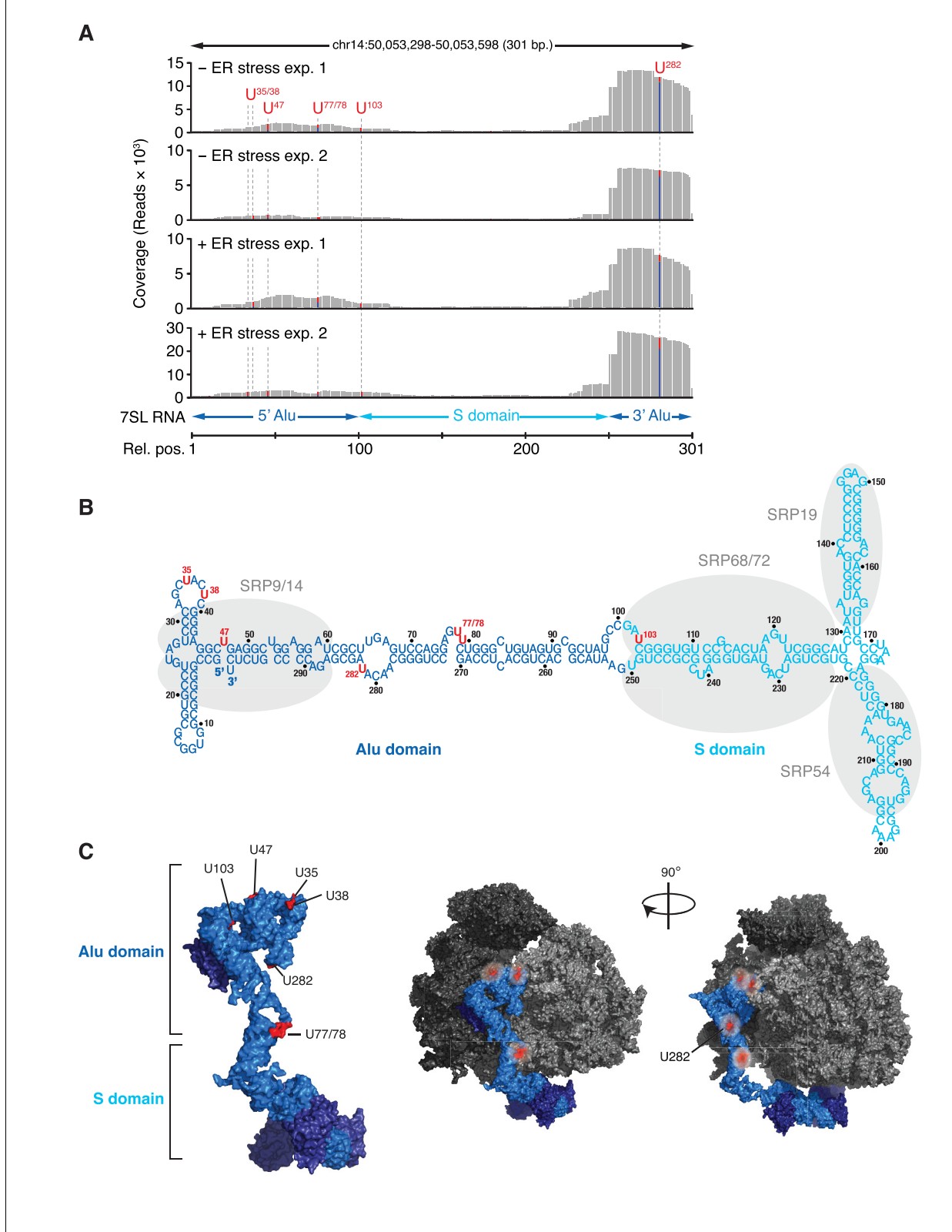

**Figure 3.** IRE1 associates with the SRP RNA. (**A**) Coverage tracks of PAR-CLIP sequencing reads in the presence or absence of chemically induced ER stress mapped to the SRP in the human genome. The colored bars indicate the proportion of reads with a T (red) or a C (blue, T→C transition), allowing mapping of the corresponding uracil on the RNA from which the reads originated (indicated over each bar). (**B**) Schematic of the SRP showing
*Figure 3 continued on next page*

*Figure 3 continued*

the crosslink sites (red circles) identified in *RN7SL* by PAR-CLIP. Shaded ovals indicate the SRP proteins. (C) Crosslink sites (red) identified by PAR-CLIP on *RN7SL* and mapped onto the structure of SRP (PDB accession number: 3JAJ). Free and ribosome-bound views of SRP are shown.

DOI: https://doi.org/10.7554/eLife.43036.008

The following figure supplement is available for figure 3:

**Figure supplement 1.** SRP RNA remains intact during ER stress.

DOI: https://doi.org/10.7554/eLife.43036.009

(*Figure 3A*), suggesting a specific IRE1-SRP association rather than association with other Alu transcripts. All crosslinked sites on SRP RNA occurred in surface accessible single-stranded regions (*Figure 3B and C*), consistent with the chemistry of the photo-crosslinking reaction.

SRP binds to the signal sequence exposed in nascent chains of translating ER-targeted mRNAs and brings these ribosomes to the ER surface (*Saraogi and Shan, 2011*; *Walter et al., 1984*). It is therefore plausible that IRE1 interacts with the SRP as part of ribosome nascent chain complexes. Supporting this notion, we identified rRNAs among the top IRE1-interacting RNAs (*Figure 1C*). We mapped PAR-CLIP reads to ribosomal DNA loci corresponding to the 45S pre-ribosomal transcript (*Figure 4A*). IRE1 preferentially associated with discrete regions in 28S rRNA and 18S rRNA (*Figure 4A*, indicated by open arrowheads), whereas we mapped substantially fewer reads to 5.8S rRNA. Considering that (i) 5.8S rRNA is processed from the same 45S precursor, (ii) all rRNAs are equimolar, and (iii) our libraries enrich for small RNAs, these results strongly suggested that the identified IRE1-ribosome associations are specific.

In addition to SRP RNA and rRNAs, we reproducibly identified a common-core set of 106 tRNAs (corresponding to ~17% of 610 annotated human tRNAs) present in the absence and presence of ER stress (*Figure 4B*, *Figure 4—figure supplement 1A*, and *Supplementary file 2*). T→C transitions occurred in the same tRNA regions (*Figure 4E* and *Figure 4—figure supplement 1B*) with a general trend towards enrichment for specific tRNAs upon chemical induction of ER stress (*Figure 4B–D* and *Figure 4—figure supplement 1C*). To assess whether there was a bias for IRE1 to interact with abundant tRNAs translating frequently used codons, we grouped the crosslinked tRNAs by anticodon. This analysis revealed that IRE1 associated with tRNAs that recognize both common and rare codons (*Figure 4C–D*, and *Figure 4—figure supplement 1C*).

IRE1β processes 28S rRNA upon ER stress to reduce global translation and thereby protein folding load (*Nakamura et al., 2011*). It is therefore plausible that IRE1α processes SRP and tRNAs to reduce the level of global translation. To test this notion, we used Northern blots to probe for SRP as well as qRT-PCR analyses with primers that detect the 3'-Alu, 5'-Alu, or S domains of SRP. We did not detect IRE1-dependent cleavage of SRP RNA by either technique even after pre-treatment of cells with 5-fluorouracil to block exosome function (*Kammler et al., 2008*) to enhance the likelihood of detection of partial cleavage intermediates (*Figure 3—figure supplement 1A and B*). These results indicated that SRP RNA is not degraded during ER stress and suggested that the crosslinks captured by the PAR-CLIP experiments reflect a structural association between SRP with IRE1. Importantly, and akin to our results on SRP RNA, we likewise detected no tRNA cleavage products (*Figure 4—figure supplement 2*).

## IRE1 interacts with a defined region on the ribosomal surface

Our data converged on a model wherein IRE1 associates with translating ribosomes on the ER surface, where it can crosslink to the SRP, ribosomal and tRNAs, as a part of ribosome-nascent chain complexes. The crosslink sites on the ribosomal RNAs as well as those found on the SRP RNA converged to a distinct region on the ribosome surface which is topologically consistent with the presence of a long linker between the transmembrane domain and the kinase/RNase (KR) domains of IRE1 (*Figure 5A–C*, and *Figure 5—figure supplement 1*). When we mapped the crosslink sites on tRNAs illustrated in *Figure 4E and F*, onto structures in which tRNAs are positioned at the A/P hybrid site on ribosomes (*Voorhees et al., 2014*), we found that IRE1 crosslink sites reside on exposed surfaces on translating ribosomes (*Figure 5D*), further strengthening our model. Notably, IRE1 crosslink sites on rRNAs did not coincide with the crosslink sites of the dead box helicase DDX3, which were identified in an earlier study (*Oh et al., 2016*), indicating that the identified IRE1-

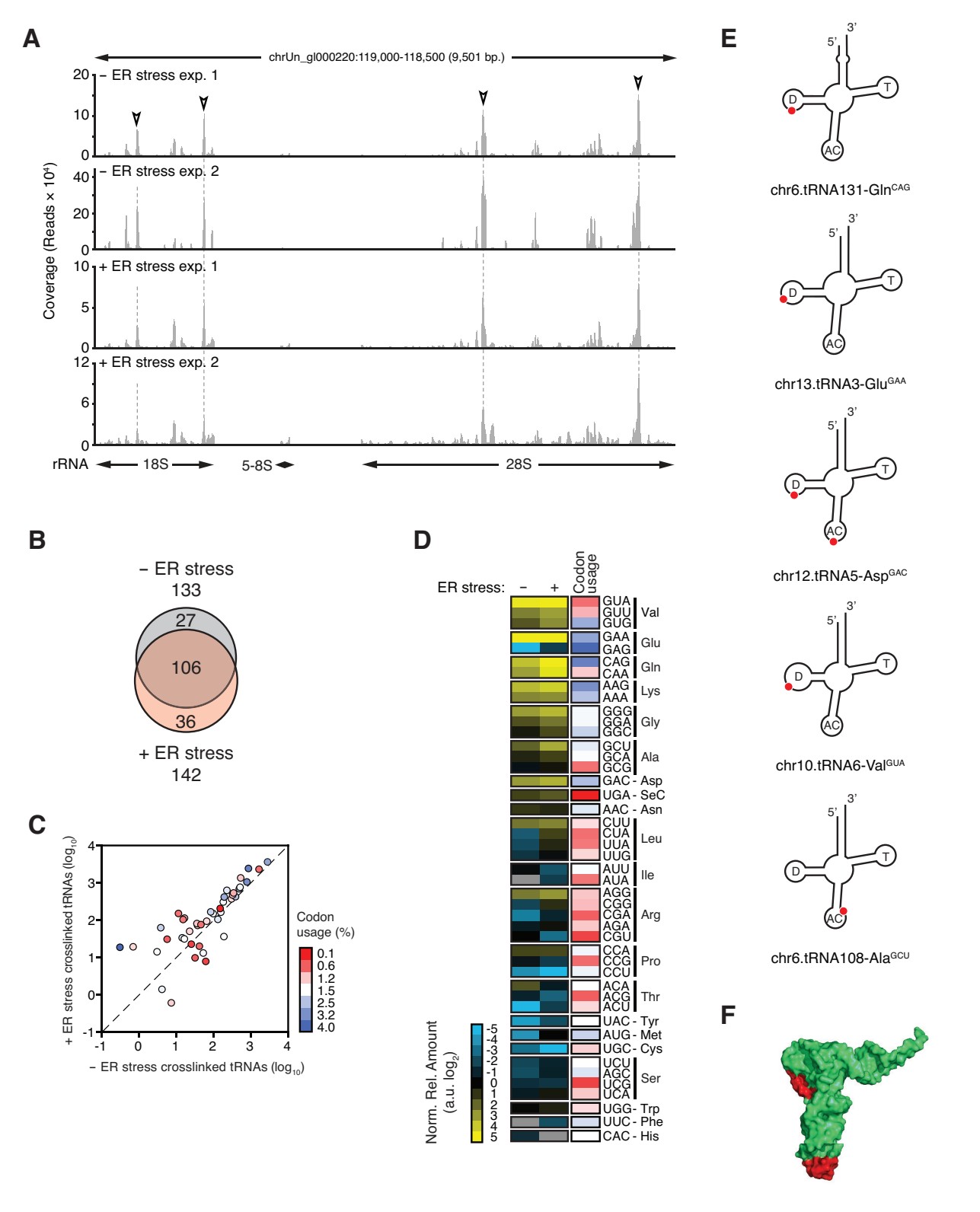

**Figure 4.** IRE1 associates with specific tRNAs. (**A**) Coverage tracks of PAR-CLIP sequencing reads recovered in the presence or absence of chemically induced ER stress mapped to the 45S pre-ribosomal loci (RNA45S5) in the human genome. Discrete regions of preferential association between IRE1 and rRNA are indicated by arrowheads. (**B**) Venn diagrams showing the numbers of crosslinked tRNAs recovered by PAR-CLIP in the presence or absence of chemically induced ER stress. (**C**) Scatter plot showing the abundance of PAR-CLIP recovered tRNAs grouped by codon-anticodon in the

*Figure 4 continued on next page*

*Figure 4 continued*
presence or absence of chemically induced ER stress (copies per million reads: sum of geometric means of copy number of each tRNA with the same anticodon in biological duplicates). Specific tRNA examples are indicated. Codons are colored based on codon usage from the codon usage database (http://www.kazusa.or.jp/codon/). The diagonal dashed line indicates a slope of 1. (D) Heat map showing the relative amounts of PAR-CLIP recovered tRNAs grouped by codon-anticodon in the presence or absence of chemically induced ER stress. Codons are colored as in panel C. (E) Schematic representation of the crosslink sites on top tRNA hits. (F) Topographical mapping of the PAR-CLIP identified crosslink sites on tRNAs.
DOI: https://doi.org/10.7554/eLife.43036.010

The following figure supplements are available for figure 4:

**Figure supplement 1.** Reproducible identification of IRE1-crosslinked tRNAs.
DOI: https://doi.org/10.7554/eLife.43036.011
**Figure supplement 2.** tRNAs remain intact during ER stress.
DOI: https://doi.org/10.7554/eLife.43036.012

ribosome interactions are indeed specific (*Figure 5—figure supplement 2A*). Taken together, our data support a model in which IRE1 associates with translating ribosomes on the ER surface.

To test this notion, we employed an orthogonal approach to identify IRE1 interacting proteins by immunoprecipitation followed by mass-spectrometry (IP-MS) from detergent extracts. Native IP-MS experiments not only confirmed a previously identified interaction between IRE1 and the Sec61 translocon (*Plumb et al., 2015*), but also validated our PAR-CLIP findings, verifying that IRE1 associates with both the SRP and the ribosome (*Figure 5F* and *Supplementary file 3*)

To further corroborate our results, we next employed a modified PAR-CLIP protocol ('PAR-CLIP-MS'), in which we identified by MS proteins that become tethered to IRE1 by crosslinked RNA fragments. For these experiments, we performed a more extensive RNase digestion after crosslinking but prior to the denaturing IP. Under such conditions, we estimate that RNA fragments were reduced to an averaged length below 20 nucleotides (*Figure 1—figure supplement 1E*), thereby minimizing longer-range interactions. We applied stringent cut-off criteria by requiring a > 1000-fold enrichment of proteins identified in the presence versus absence of UV crosslinking. Pairwise comparisons of the native IRE1-protein interactions with those obtained by PAR-CLIP-MS shows that IRE1 crosslinks via an RNA tether to specific ribosomal proteins (RLP9, RPL10A, RPS14, RPS24) with an amazing $10^6$ discriminatory power (*Figure 5F and G* and *Supplementary file 3*). Notably, the set of identified proteins also included the RTCB tRNA ligase, which is a principal component required for the ligation of the *XBP1* mRNA exons generated by IRE1 (*Kosmaczewski et al., 2014*; *Lu et al., 2014*), as well as SSR2, an ER translocon-associated protein (TRAP) component (*Figure 5G*). We validated these interactions for individual proteins by IP-immunoblot analysis (*Figure 5H*), lending support to the robustness of our methods.

Interestingly, two of the four ribosomal proteins identified by PAR-CLIP-MS (RPL9, RPS24), the mapped SRP RNA crosslinks, tRNA crosslinks, and rRNA-IRE1 crosslinks identified by PAR-CLIP (*Figure 5B*), all converge on a defined region spanning from the aminoacyl tRNA-binding site (A-site) extending to the ribosomal exit tunnel on the ribosome surface. The remaining two ribosomal proteins (RPL10A, RPS14) map in the vicinity of the mRNA exit site in the ribosome and are likely linked to IRE1 via a short mRNA fragment (*Figure 5—figure supplement 2B*). Taken together, these results suggest that IRE1 interacts with the ribosome and co-translational protein targeting machinery directly.

## IRE1 interacts with ribosomes *in vitro*

To test this notion, we assessed IRE1-KR binding to purified mammalian ribosomes using two different biochemical approaches. First by thermophoresis, we measured IRE1-KR's affinity for ribosomes by monitoring the changes in the mobility of fluorescently labeled recombinant cytosolic IRE1-KR in the presence of purified ribosomes (*Figure 6A*). These experiments revealed that IRE1-KR bound ribosomes with high affinity ($K_d = 30 \pm 11$ nM). These results were corroborated by co-sedimentation assays in which we recovered IRE1-KR in the pellet fraction after adding increasing amounts of purified ribosomes (*Figure 6B*). Importantly, IRE1-KR did not co-sediment with dissociated 40S or 60S ribosomal subunits (*Figure 6C*), suggesting that IRE1 preferentially selects intact ribosomes. This is consistent with the mapping data in *Figure 5* showing that the IRE1 interaction region extends between the two ribosomal subunits. In agreement with these results, size exclusion chromatography

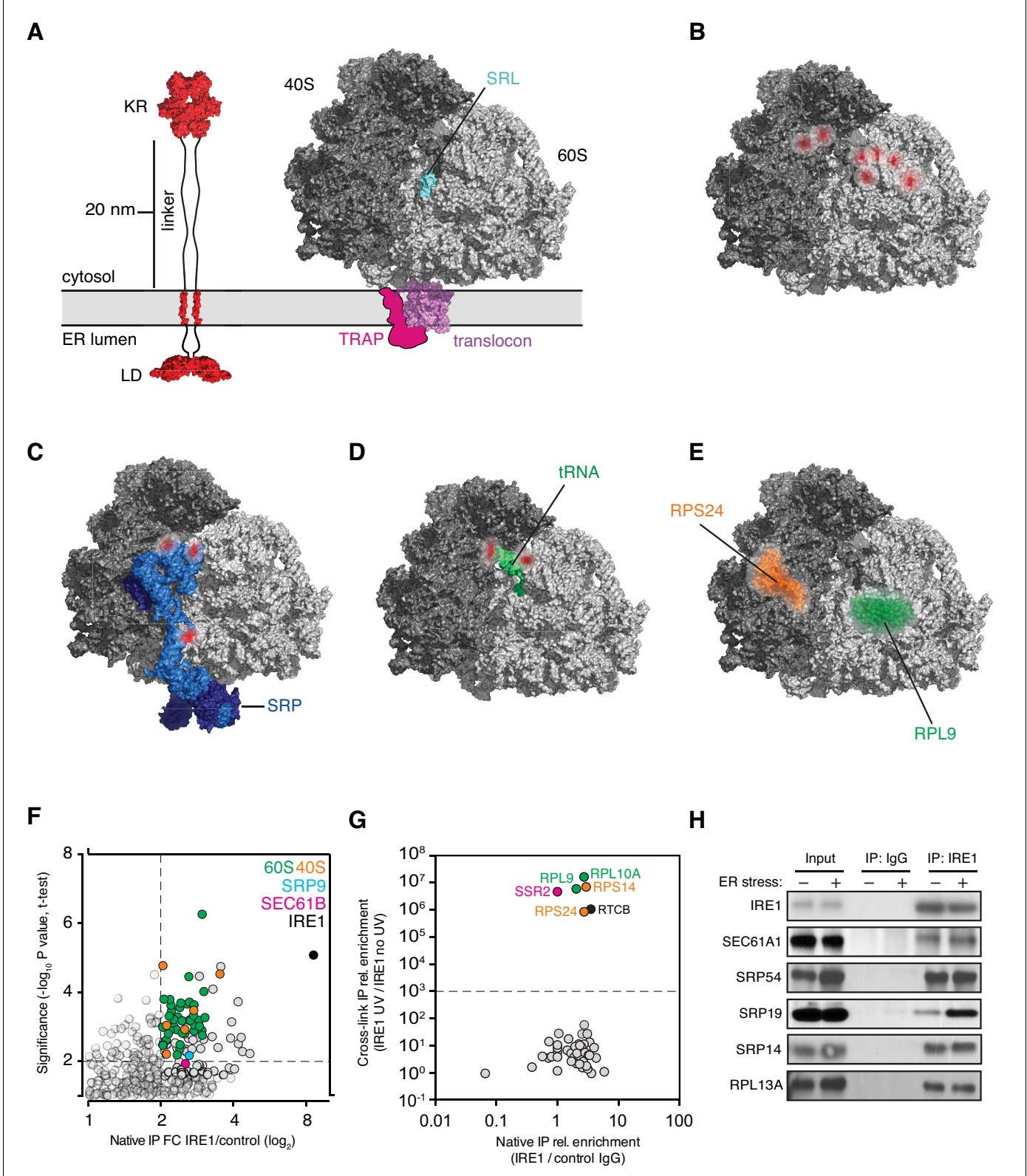

**Figure 5.** IRE1 and ribosome interact tightly over a well-defined region. (A) Depiction of the topologies of IRE1 and an ER-bound ribosome docked on the translocon (after PDB accession number 3J7R). The linker tethering IRE1's transmembrane and kinase/endonuclease domains is drawn to scale assuming an extended conformation. SRL: Sarcin-ricin loop. (B) Topographical localization of the IRE1 crosslink sites on ribosomes identified in PAR-CLIP experiments. Crosslinked regions in ribosomal RNAs (from *Figure 4A*) are indicated in red. The sarcin-ricin loop in the 60S subunit (colored in

*Figure 5 continued*

cyan) is shown as a topographical reference. Based on published structures with PDB entry 3J7R. (C) Topographical localization of the IRE1 crosslink sites on SRP engaged with the ribosome (PDB accession number 3JAJ). (D) Topographical mapping of the PAR-CLIP identified crosslink sites on tRNAs positioned in the A/P site of a ribosome. Based on PDB accession number 3J7R. (E) Topographical localization of ribosomal proteins crosslinked to IRE1 through RNA bridges and recovered in denaturing IPs. Note that the sites of IRE1-ribosome interaction coalesce in a band stretching from the A-site to the vicinity of the translocon. (F) Scatter plot showing hits obtained in native IRE1 IPs followed by LC/MS analysis of interacting IRE1 protein partners. Relevant hits are indicated. (G) Scatter plot comparing the relative abundance of ribosomal proteins and the relevant proteins SSR2 and RTCB recovered in native IPs and denaturing IPs. Each dot is a single ribosomal protein. Specific proteins recovered in denaturing IPs are indicated. (H) IP-Western blot validations of select IRE1-SRP and IRE1-ribosome interactions.

DOI: https://doi.org/10.7554/eLife.43036.013

The following figure supplements are available for figure 5:

**Figure supplement 1.** IRE1 crosslink sites mapped onto the secondary structure of the large and small ribosomal subunit RNAs (*Petrov et al., 2014*).
DOI: https://doi.org/10.7554/eLife.43036.014
**Figure supplement 2.** IRE1 associates with specific surfaces on the ribosome.
DOI: https://doi.org/10.7554/eLife.43036.015

experiments revealed that IRE1-KR eluted in earlier fractions when incubated with ribosomes (*Figure 6D*). To determine kinetic properties of the interaction between IRE1 and ribosomes, we next used biolayer interferometry. These experiments revealed that IRE1-KR binds to ribosomes with fast association ($k_{on}$ = 1.81 ± 0.12 x $10^6$ $M^{-1}$ $s^{-1}$) and dissociation ($k_{off}$ = 5.98 ± 0.13 x $10^{-2}$ $s^{-1}$) kinetics, with a similar dissociation constant ($K_d$ = 33.4 ± 2.3 nM) as that measured by thermophoresis (*Figure 6A*). Again, IRE1-KR did not show binding to 60S subunits in these experiments (*Figure 6—figure supplement 1A*). Taken together, these experiments revealed that IRE1-KR associates with 80S ribosomes *in vitro*.

To test whether ribosome binding impacts IRE1's RNase activity, we measured the effect of ribosome binding on the *in vitro* cleavage efficiency of two IRE1 substrates: a single RNA hairpin, HP21, which is a 21-nucleotide canonical IRE1 RNA substrate derived from the *XBP1* mRNA, as well as a double hairpin derived from *XBP1* mRNA. The presence of ribosomes slightly enhanced the catalytic activity of IRE1-KR in different ribosome concentration regimes (*Figure 6F* and *Figure 6—figure supplement 1B*). We did not observe cleavage of rRNAs, consistent with previously reported observations (*Nakamura et al., 2011*). Therefore, binding to ribosomes does not substantially impact IRE1 RNase activity nor does it compromise the integrity of the ribosome.

## Peptide stretches encoded in IRE1-interacting mRNAs bind IRE1's lumenal domain

Our observations that IRE1 preferentially associated with ER-bound mRNAs and ribosomes suggested that IRE1 might co-translationally survey ER client protein entry. Such control could be accomplished through interactions between IRE1's lumenal sensor domain and nascent polypeptides entering the ER lumen encoded in such mRNAs. To begin gathering support for this notion, we designed a peptide array tiling 18-mer peptides derived from proteins encoded by top hit PAR-CLIP mRNAs. Incubation of this array with recombinantly expressed IRE1 lumenal domain fused to maltose binding protein indicated that IRE1's sensor domain bound to distinct sequences embedded in ER lumenal regions in those proteins (*Figure 6G and H*). These data lend support to a mechanism wherein IRE1 surveys the folding status of client proteins entering the ER lumen as they are being synthesized.

## Discussion

We identified RNAs and RNA/protein complexes that interact with IRE1 in living cells using three unbiased and complementary approaches. First, we used PAR-CLIP to catalog RNAs that associate with IRE1. This method provided us with high structural resolution due to the single nucleotide-precision with which PAR-CLIP crosslinking sites are identified. Second, we used PAR-CLIP-MS to identify proteins bridged IRE1 by short RNA fragments, allowing us to identify ribonucleoprotein complexes. Third, we used IP-MS to identify components that associate stably with IRE1 under native conditions. Together, our results portray a detailed picture of the IRE1 interactome in living cells that is far more

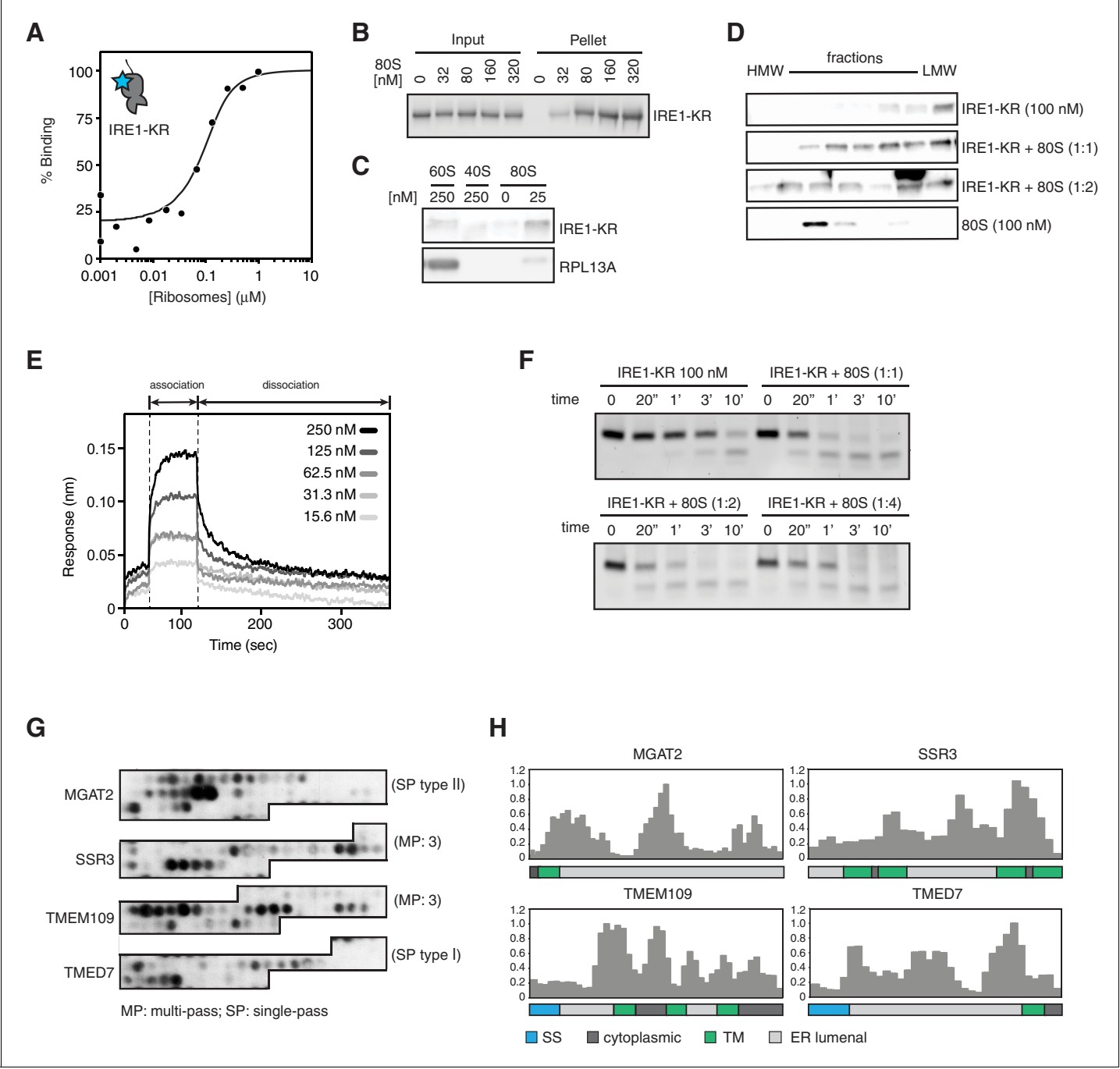

**Figure 6.** IRE1 associates with ribosomes and peptide stretches encoded in specific mRNAs. (**A**) Thermophoresis analysis of IRE1-ribosome binding affinity. A fluorescently labeled cytosolic portion of IRE1 consisting of its kinase and nuclease domains preceded by a 43-residue portion of the linker that tethers the C-terminal kinase/nuclease domains to the ER membrane were employed to assess its binding affinity to purified ribosomes. (**B-C**) IRE1-KR-ribosome interactions validated by co-sedimentation assays. Immunoblots of the pellets of the sucrose cushions at various ribosome concentrations separated by SDS-PAGE probed with the indicated antibodies. (**D**) Size exclusion chromatography fractions analyzed by immunoblot showing that IRE1-KR elutes earlier from a Sephacryl-300 column upon binding to ribosomes. The fractions were separated by SDS-PAGE and probed with the indicated antibodies against IRE1 or the ribosomal protein RPL13A. (**E**) Bio-layer interferometry (BLI) signal of IRE1-KR loaded biosensors dipped into a solution containing ribosomes at the indicated concentrations for the indicated time. The increase in the BLI signal indicates association between IRE1 and ribosomes. Dipping the IRE1-KR coupled sensors into buffer devoid of ribosomes promotes IRE1-ribosome dissociation and the consequent drop in the BLI signal (**F**) SYBR-gold stained TBE-Urea PAGE gels showing the effects of the presence of purified ribosomes in an *in vitro* cleavage assay for IRE1-KR employing a minimal 21-mer archetypal substrate RNA corresponding to the 3' canonical hairpin structure found in the bifurcated stem-loop of
*Figure 6 continued on next page*

*Figure 6 continued*

*XBP1* mRNA of human origin (**G**) Peptide array immunoblot of 18-mers spotted tiled along proteins encoded by select PAR-CLIP target mRNAs. (**H**) Quantification of array in G. The schematics illustrate the topology of the proteins encoded by the transcripts.

DOI: https://doi.org/10.7554/eLife.43036.016

The following figure supplement is available for figure 6:

**Figure supplement 1.** IRE1-KR does not associate with the 60S ribosomal subunit.

DOI: https://doi.org/10.7554/eLife.43036.017

complex than anticipated. In particular, IRE1 interactome analysis suggests IRE1's intimate engagement with the protein targeting and translation machineries. This notion is supported by the following lines of evidence: First and most surprisingly, we found that IRE1 associates with structural, non-coding RNAs that include ribosomal RNAs, SRP RNA, and select tRNAs. Second, IRE1 interacts with several components of the co-translational protein targeting and translocation machinery that include the translocon, the translocon-associated component TRAP, SRP proteins, and ribosomal proteins. Importantly, the crosslink sites between IRE1 and the structural RNAs identified here, as well as those between IRE1 and RNA-bridged proteins, converge on a discrete region on the ribosome surface, spanning from the ribosomal A-site to the nascent chain exit tunnel. Third, IRE1 associates with a select population of mRNAs, strongly enriched in ER-bound mRNAs those that encode proteins that traverse or reside in the secretory pathway and IRE1's ER-lumenal domain engages polypeptide regions encoded by these mRNAs. Finally, the cytosolic domains of IRE1 tightly bind 80S ribosomes *in vitro*.

Our data support a model wherein the UPR and co-translational targeting to the ER are linked. According to this view, ribosome binding to SRP or to the translocon represents two distinct stages during co-translational targeting in which IRE1 may engage with ER-targeted mRNAs as depicted in *Figure 7*. First, in a 'preemptive' mode, IRE1 contacts SRP in the context of ribosomes that have arrived at the surface of the ER but have not yet engaged with the translocon. Second, in a 'surveillance' mode, IRE1 contacts the ribosome after engagement with the translocon is complete and protein translocation has initiated. In this view, the two postulated modes of IRE1 engagement exert different regulatory functions: The preemptive mode would enable IRE1-mediated degradation of newly targeted mRNAs. Such a situation may arise, for example, under conditions of ER stress when IRE1 is widely activated in the ER lumen and thus serves to prevent further increase of the ER protein-folding load. By contrast, the surveillance mode would enable IRE1-mediated degradation of

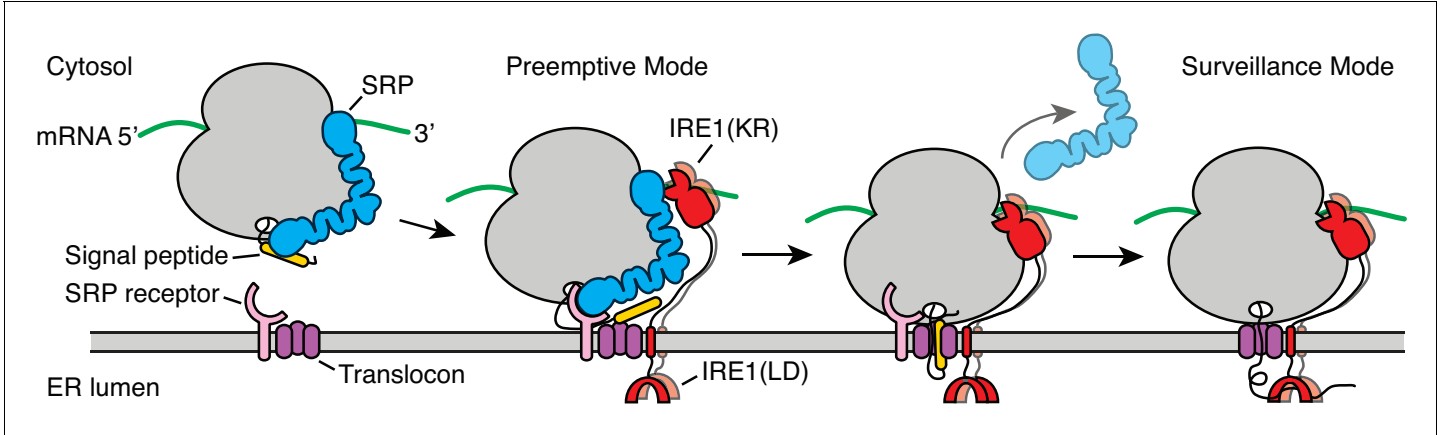

**Figure 7.** Model for co-translational IRE1 engagement. The linker tethering IRE1's transmembrane domain and its kinase/nuclease domains (KR) is drawn to scale as an extended chain. LD: lumenal domain.

DOI: https://doi.org/10.7554/eLife.43036.018

mRNAs encoding protein chains that have difficulty folding in the ER. In this way, IRE1 activation would be local and private, restricted to the immediate vicinity of the translocon that is in the process of injecting a potentially problematic nascent protein into the ER lumen. As such, IRE1 would exercise coincidence detection of features in the nascent chain and the mRNA, as previously suggested (*Hollien and Weissman, 2006*). This mechanism could provide the means to identify the mRNA on the cytosolic face of the ER when the translocation and the folding rates are not perfectly coupled and may explain why certain mRNAs are preferentially selected for RIDD.

The strong bias towards mRNAs encoding transmembrane proteins among the core-set of IRE1-interacting mRNAs substantiates the notion that IRE1 surveys the entry of select clients/residents into the ER lumen. Moreover, the prevalence of rare-codon tRNAs bound by IRE1 further supports the idea that IRE1 engages specific substrates that could pose a folding challenge and therefore evolved lower localized translation rates (*Pechmann and Frydman, 2013*; *Pechmann et al., 2013*).

The model shown in *Figure 7* is consistent with previous findings that have suggested co-translational IRE1 engagement, including the observation that SRP targets paused ribosomes translating unspliced *XBP1* mRNA to the ER membrane (*Yanagitani et al., 2009*; *Yanagitani et al., 2011*) and the identification of a specific site on IRE1's lumenal domain that provides a direct point of contact to the translocon (*Plumb et al., 2015*). Moreover, recent work showed that the genetic depletion of translocon components selectively induces the IRE1 branch of the UPR, suggesting specific mechanisms of control linking IRE1 signaling and the co-translational insertion of proteins into the ER lumen (*Adamson et al., 2016*).

## Methodological considerations

Applying PAR-CLIP-based approaches to map the IRE1 interactome posed technical challenges. While powerful, PAR-CLIP relies on enrichment by immunoprecipitation and target protein abundance is a major determinant for the success of the method. IRE1 is not an abundant protein, and, as an enzyme, its engagement with substrate RNAs is transient, and often rapid, for example during *XBP1* mRNA splicing (*Peschek et al., 2015*). For these reasons, the detection of IRE1-bound RNAs by PAR-CLIP required overexpression of IRE1 from an inducible promoter. Overexpression of IRE1 leads to constitutive, albeit still only partial activation in the absence of ER stress inducers, thus preventing us from capturing IRE1-RNA interactions in its inactive state. We partially overcame these experimental limitations by activating IRE1 further (*e.g.*, beyond the elevated baseline) with chemical ER stress inducers as one of our experimental conditions and applying stringent filters in categorizing RNAs as IRE1-interactors. Such strategy allowed us to pinpoint core-sets of IRE1-interacting RNAs.

The IRE1-dependent cleavage of the mRNAs we identified in this study was either very low or undetectable in cells and inefficient when reconstituted *in vitro*, suggesting that they are suboptimal substrates. This difference in cleavage efficiency when compared to known RIDD or splicing substrates is likely to result from intrinsic RNA features that modulate cleavage. Ideal IRE1 substrates harboring the archetypal 7-mer loop substrate hairpins with a GC(A/C)G central motif found in *XBP1* mRNAs (*Peschek et al., 2015*) and in the canonical RIDD substrate *BLOC1S1* mRNA (*Moore and Hollien, 2015*) are cleaved by IRE1 with fast kinetics. Indeed, the canonical structures recognized by IRE1 in the *XBP1* mRNA, as well as select RIDD substrates including *BLOC1S1* mRNA, are cleaved fast *in vitro* (*Li et al., 2018*; *Peschek et al., 2015*; and *Figure 2D*). In line with this view, we did not find the *XBP1* or *BLOC1S1* mRNAs among our strong PAR-CLIP hits, which we can attribute to RNA-IRE1 dwell times that are too short for efficient enough photo-crosslinking that would allow detection.

## Considerations on the potential for regulation of protein synthesis by IRE1

The RNA-protein and protein-protein interactions identified here place IRE1 at the crossroads of protein synthesis and quality control at the ER membrane, setting appropriate mRNA lifetimes and fine-tuning protein production. The importance of controlling protein synthesis during stress is paralleled by several orthogonal mechanisms. For instance, the kinases that activate the integrated stress response attenuate translation (*Harding et al., 2003*; *Pakos-Zebrucka et al., 2016*), and protein synthesis can be halted by cleavage of tRNAs by angiogenin (*Fu et al., 2009*; *Ivanov et al., 2011*;

*Yamasaki et al., 2009*). By contrast to the readily detectable tRNA cleavage by angiogenin, we did not detect cleavage of tRNAs or SRP RNA under the conditions used here, although we cannot exclude that IRE1 cleaves a small fraction of structural RNAs that engage with it. In line with our observations, a recent report shows that IRE1's sister protein, the pseudokinase-endoribonuclease RNaseL, a central effector of the innate immune system, also associates with translating ribosomes (*Nogimori et al., 2018*). Together with this observation, our results suggest a common specialized surveillance mechanism in the UPR and in the type I interferon response that controls the half-lives of endogenous and exogenous RNAs. Such regulation may serve to curtail threats to cellular homeostasis, such as excessive protein synthesis during limiting protein folding conditions or translation of viral proteins, respectively.

## Stoichiometric considerations

Back-of-the-envelope calculations indicate that the abundance of translocons in the ER membrane is at least 100-fold more than that of IRE1. Thus, IRE1 can only engage a subset of translocons and/or polysomes at any given time. The specific associations we observed for less than a hundred mRNAs that encode proteins residing in or traversing the secretory pathway may reflect that these mRNAs encode potentially problematic ER clients/residents. For example, one of the most robust PAR-CLIP hits was the mRNA encoding TMEM109, a small multi-pass transmembrane protein that forms very large homo-oligomeric assemblies and may be difficult to assemble (*Venturi et al., 2011*).

The concept of co-translational engagement of IRE1 raises numerous questions that may necessitate the re-evaluation of current models. For example, IRE1 oligomers visualized as dynamic foci in UPR-activated cells are widely assumed to be active 'splicing factories' (*Aragón et al., 2009*; *Li et al., 2010*) (*i.e.*, constitute the sites where *XBP1* mRNA splicing takes place). A possibility is that IRE1 foci form a reservoir from which IRE1 molecules are recruited to membrane-engaged ribosome/nascent chain complexes, reconciling foci with the far more diffuse distribution of membrane-bound polysomes on the ER surface. Alternatively, IRE1 foci may represent a different state in the IRE1 signaling chain; for example, foci could embody sites for IRE1 degradation or recycling. Notwithstanding the many open questions that remain, our results strongly converge on the intriguing possibility of synchronous monitoring of the folding status in the ER lumen.

## Materials and methods

### Cell line, cell culture and drugs

HEK293Trex cells (Thermo Fisher Scientific) harboring an epitope-tagged allele of IRE1 (3× FLAG hexahistidine) integrated at a single genomic locus and driven of a tetracycline inducible promoter were generated as described (*Li et al., 2010*). The cells were grown in high-glucose DMEM supplemented with 10% tetracycline-free certified fetal bovine serum (FBS) (Clontech), 4 mM L-glutamine and penicillin/streptomycin. RPMI-8226 multiple myeloma cells were purchased from the German Collection of Microorganisms and Cell Cultures (DSMZ). The cells were grown in RPMI1640 medium supplemented with 10% heat-inactivated FBS, 4 mM L-glutamine and penicillin/streptomycin. Cells were kept at 37°C, 5% $CO_2$ until harvesting for experiments. The following drugs were used at the concentrations and times noted below or elsewhere: 4µ8C (Matrix Chemicals), 4-thiouridine (Sigma-Aldrich), 5-fluorouracil (Sigma-Aldrich), actinomycin D (Sigma-Aldrich), doxycycline hyclate (Sigma-Aldrich), thapsigargin (Sigma-Aldrich), tunicamycin (EMD Millipore). The cells tested negative for mycoplasma contamination.

### PAR-CLIP

All PAR-CLIP experiments were performed as biological duplicates following the protocol below.

#### Cell culture, drug treatments and photo-crosslinking

HEK293Trex cells were seeded at 8 million cells per plate in ø15 cm tissue culture plates. 15 × ø15 cm plates were used for each condition in each experiment. 24 hr after seeding, doxycycline was added to a final concentration of 2.50–3.33 nM to induce expression of epitope-tagged IRE1. 6 hr after addition of doxycycline the cells were treated with 100 µM 4-thiouridine and incubated for an additional 12 hr. Then, 5 µg/mL tunicamycin were added and the cells were incubated for an

additional 4 hr to generate high-level ER stress. Control cells were incubated with an equal amount of DMSO. After incubation the medium was removed and the cells were rinsed twice with cold PBS. The plates were placed on a bed of ice and the cells were irradiated in a 365 nM UV light crosslinker (Spectronics Corp.) with 150 mJ/cm$^2$ (approx. 45 sec). The cells were scraped off the plates and collected. The approximate yield (biomass) per condition, per experiment, was ~1 mL wet cell pellet. The cell pellets were flash-frozen in liquid nitrogen and stored at -80°C until use.

## Cell lysis, subcellular fractionation, and partial RNase digestion

Cell pellets were thawed on ice and washed once with three pellet volumes of ice-cold PBS. The cells were collected by centrifugation at 500 × $g$ at 4°C for 5 min and re-suspended in three pellet volumes of ice-cold lysis buffer 1 (25 mM HEPES pH 7.5, 150 mM NaCl, 1 mM EDTA, 300 µg/mL digitonin (Promega),1 unit/µL RNasin (Promega) and Halt EDTA-free protease inhibitor cocktail (Thermo Fisher Scientific), and incubated 10 min on ice with gentle vortex every 2–3 min. The digitonin-permeabilized cells were collected by centrifugation at 500 × $g$ at 4°C for 5 min and the cytosolic fraction (supernatant) was removed. The pellets were washed by re-suspension in three pellet volumes of ice-cold lysis buffer 2 (25 mM HEPES pH 7.5, 150 mM NaCl, 1 mM EDTA, 0.5 unit/mL RNasin and Halt EDTA-free protease inhibitor cocktail), and the permeabilized cells were collected by centrifugation at 500 × $g$ at 4°C for 5 min. To extract the membranous organelle fraction, the pellets were re-suspended in three pellet volumes of ice-cold lysis buffer 3 (25 mM HEPES pH 7.5, 150 mM NaCl, 1 mM EDTA, 10% glycerol, 1% Triton X-100, 1 unit/mL RNasin and Halt EDTA-free protease inhibitor cocktail) and incubated on ice for 10 min with gentle vortex every 2–3 min. The lysates were centrifuged at 3,800 × $g$ at 4°C for 5 min to pellet nuclei and cell debris. The soluble membrane fraction was then transferred to pre-chilled RNase-free non-stick microcentrifuge tubes (Thermo Fisher Scientific) and the volume was adjusted to 758 µL with excess lysis buffer 3. Approximately 2 µL of an RNase T1 (Thermo Fisher Scientific) dilution in lysis buffer 3 was then added to the membranous fractions so that the final concentration of RNaseT1 was 0.25–0.3 unit/µL and the volume was approximately 800 µL. To digest the RNA, the lysates were incubated at 25°C for 30 min.

## Denaturing immunoprecipitation, end-repair and radiolabeling of crosslinked RNA tags

To denature the lysates, ~40 µL of 20% SDS, corresponding to 1/19 of the volume, were added to the RNase-digested lysates, the tubes were briefly vortexed, and were incubated at 95°C for 6–7 min. The tubes were then placed immediately on ice and incubated for an additional 3 min. The digested, denatured lysates were cleared by centrifugation at 21,000 × $g$ at 4°C for 15 min in a tabletop microcentrifuge, and the supernatant was transferred to RNase-free pre-chilled 15 mL conical tubes. 9 volumes of IP buffer (25 mM HEPES pH 7.5, 150 mM NaCl, 1 mM EDTA, 10% glycerol, 1% Triton X-100, 0.5% sodium deoxycholate, 1 unit/mL RNasin and Halt EDTA-free protease inhibitor cocktail) were added to each tube to dilute the SDS concentration to 0.1% before immunoprecipitation (IP). For each IP, 50 µg anti-FLAG M2 antibody (Sigma-Aldrich) were pre-bound to 500 µL of magnetic Dynabeads Protein G solution (Thermo Fisher Scientific) by incubation for 45 min at ~25°C (room-temperature) in a rotating platform. The antibody-bound beads were washed twice with IP buffer without RNasin and supplemented with 0.1% SDS. The beads were re-suspended in their original volume in IP buffer supplemented with 0.1% SDS and 1 unit/µL RNasin, and were added to the diluted lysates. Immune complexes were allowed to form by incubation at 4°C for 3 hr on a rotating platform. The bead-bound immunoprecipitates from each 15 mL conical tube were pooled into a single RNAse-free non-stick microcentrifuge tube and were washed 4 times for 5 min each time at 4°C with ice-cold HEPES-buffered high-salt wash buffer (25 mM HEPES pH 7.5, 1M NaCl, 1 mM EDTA, 1% Triton X-100, 0.5% sodium deoxycholate, 0.1% SDS, 1 unit/mL RNasin and Halt EDTA-free protease inhibitor cocktail). The beads were rinsed twice with ice-cold calf intestinal alkaline phosphatase (CIP) buffer (50 mM Tris pH 7.9, 100 mM NaCl, 10 mM MgCl$_2$, 0.2% Triton X-100 and 1 mM DTT) and re-suspended in 111 µL CIP buffer. 60 units CIP (New England Biolabs) and 120 units RNasin were added and the beads were incubated for 10 min at 37°C. The beads were rinsed twice with ice-cold CIP wash buffer (50 mM Tris pH 7.5, 20 mM EGTA, 0.5% Triton X-100), and two more times with ice-cold polynucleotide kinase (PNK) buffer (50 mM Tris pH 7.5, 50 mM NaCl, 10 mM MgCl$_2$, 0.2% Triton X-100), and were re-suspended in 105 µL ice-cold PNK buffer

supplemented with 1 mM DTT. To radiolabel the 5' termini of the IRE1-crosslinked RNAs, 60 µCi of γ-$^{32}$P-ATP (3,000 Ci/mmol, 10 mCi/ml; Perkin Elmer), 60 units of T4 PNK (New England Biolabs) and 120 units RNasin were added and the immunoprecipitates were incubated for 25 min at 37°C. The labeling reactions were terminated by addition of cold ATP to a final concentration of 1 mM and incubation at 37°C for an additional 5 min. The beads were rinsed four times with ice-cold Tris-buffered high-salt wash buffer (25 mM Tris pH 7.5, 1M NaCl, 1 mM EDTA, 1% Triton X-100, 0.5% sodium deoxycholate, 0.1% SDS) and re-suspended in 30 µL elution buffer (50 mM Tris pH 7.0, 4 mM EDTA, 2% SDS and 100 mM DTT). One volume of 2× SDS PAGE sample buffer (66 mM Tris pH 6.8, 2% SDS, 25% glycerol, 0.01% bromophenol blue) was added, and the samples were incubated at 70°C, shaking, for 15 min. The beads were spun-down by brief centrifugation at 6,000 × g for 30 sec in a tabletop microcentrifuge and the eluates were recovered.

## Recovery of crosslinked RNA tags

The radiolabelled eluates were separated in 4–12% NuPAGE Novex Bis-Tris gels (Thermo Fisher Scientific) using MOPS-SDS running buffer (Thermo Fisher Scientific). The gels were wrapped in plastic film and exposed to a blank phosphorscreen for 1 hr to collect autoradiograms. The radioactive bands corresponding the approximate molecular weight of IRE1 (~130 kDa) and the diffuse radioactive band above it were excised from the gels, and the gel slices were placed in a water-pre-equilibrated D-tube dialyzer (EMD Millipore) with a molecular weight cut-off of 3.5 kDa. 750 µL of 1× MOPS SDS buffer were added to each dialyzer. The dialyzers were placed immersed on a horizontal electrophoresis apparatus filled with 1× MOPS SDS buffer, and a constant voltage of 100V was applied for 2.5 hr to electroelute the crosslinked protein-RNA complexes. The electroeluates were transferred to clean RNase-free non-stick microcentrifuge tubes and 1 volume of 2× proteinase K buffer (100 mM Tris pH 7.5, 100 mM NaCl, 20 mM EDTA, 2% SDS) and RNA-grade proteinase K (Thermo Fisher Scientific) to a final concentration of 1.2 mg/mL, were added to digest the protein by incubation for 30 min at 55°C. The crosslinked RNA tags were extracted with 1 volume of acid phenol:chloroform (125:24:1) using PhaseLock Heavy gel Tubes (5 Prime). A second extraction with chloroform was performed to eliminate traces of phenol. The purified crosslinked RNA tags were precipitated with 300 mM NaOAc pH 5.5, 1 volume of ice-cold isopropanol and 10 µg Glycoblue (Thermo Fisher Scientific) overnight at -80°C. RNA pellets were recovered after centrifugation at 21,000 × g for 30 min at 4°C in a tabletop microcentrifuge. The pellets were washed with 1 mL 80% ice-cold ethanol, air-dried and re-suspended 6 µL of RNase-free water.

## Preparation of small RNA cDNA libraries for deep sequencing

1 µL of RA3 RNA 3' adapter (5'-TGGAATTCTCGGGTGCCAAGG-3') from the TruSeq RNA small Sample Prep Kit (Illumina) was added to each of the purified RNA samples above, and the mixture was placed at 80°C for 2 min, then placed immediately on ice for another 2 min. 1.5 µL of 10× T4 RNA ligase reaction buffer (500 mM Tris pH 7.5, 100 mM MgCl$_2$, 10 mM DTT; New England Biolabs) and 4.5 µL 50% PEG 8,000 (New England Biolabs), 1 µL RNase inhibitor (Illumina), and 1 µL (200 units) T4 RNA ligase 2, truncated R55K, K227Q (KQ) mutant (New England Biolabs) were added to each sample and the reactions were incubated at 16°C overnight. 12–16 hr later, 0.5 µL of 10× T4 RNA ligase reaction buffer, 1.5 µL of 50% PEG 8,000, 2 µL of RNase-free water and 1 µL (200 units) T4 RNA ligase 2 truncated KQ were added to each sample and the reactions were incubated an additional 2 hr at 25°C. The 3'-adapter-ligated RNA tags were purified by PAGE using 10% TBE-urea gels. The gels were wrapped in plastic film and exposed to a blank phosphorscreen for 2 hr to collect autoradiograms. The gel portion containing the smear of 3'-adapter-ligated RNA tags, typically spanning from ~30-100 nucleotides, was excised from the gels and placed in 0.5 mL RNase-free microcentrifuge tubes in which a small hole was pierced at the bottom with a syringe needle. The tubes containing the gel slices were nested in 1.5 mL RNase-free non-stick microcentrifuge tubes and centrifuged at 21,000 × g for 3 min to force-crush the gel through the holes. 3 volumes of RNase-free water were added to the crushed gel fragments and the slurry was incubated at 70°C, shaking, for 15 min to extract the 3'-ligated RNA tags. The slurries were transferred to Spin-X 0.45 µm tube filters (Corning Costar) and centrifuged at 21,000 × g for 3 min to remove the gel fragments. The filtrates were transferred to fresh RNase-free non-stick microcentrifuge tubes and the 3'-adapter-ligated RNA tags were precipitated with 300 mM NaOAc pH 5.5, 1 volume of ice-cold

isopropanol, and 15 µg Glycoblue for 60 min at -80°C. RNA pellets were recovered by centrifugation at 21,000 × g for 30 min at 4°C in a tabletop microcentrifuge. The pellets were washed with 0.75 mL 80% ice-cold ethanol, air-dried and re-suspended in 4 µL of RNase-free water. 1 µL of RA5 RNA 3' adapter (5'-GUUCAGAGUUCUACAGUCCGACGAUC-3') from the TruSeq RNA small Sample Prep Kit was added to each of the 3'-adapter-ligated RNA tags and then the mixture was placed at 80°C for 2 min, then placed immediately on ice for another 2 min. 2 µL of HML ligation buffer, 1 µL of RNase inhibitor and 1 µL of T4 RNA ligase, all from the TruSeq RNA small Sample Prep Kit, and 1 µL of 10 mM ATP were added to each sample and the reactions were incubated for 2 hr at 28°C. Next, 1 µL of RTP RNA reverse transcription primer (5'-GCCTTGGCACCCGAGAATTCCA-3') from the Tru-Seq RNA small Sample Prep Kit was added to each of the 5'- and 3'-adapter-ligated RNA tags, and the reactions were incubated at 70°C for 2 min and then placed immediately on ice. To prepare the cDNA libraries for deep sequencing, the primed, adapter-ligated RNA tags were reverse transcribed by adding 4 µL of 5× First Strand Buffer (Thermo Fisher Scientific), 1 µL 10 mM dNTPs (Thermo Fisher Scientific), 100 mM DTT, 1 µL RNaseOUT recombinant RNase inhibitor (Thermo Fisher Scientific), 1 µL SuperScript III (Thermo Fisher Scientific) and 1 µL RNase-free water, and the reactions were incubated at 50°C for 1 hr. The reactions were terminated by incubation at 70°C for an additional 15 min. To remove the RNA and the radioactive label, 1 µL of RNase H (2 units/uL, Thermo Fisher Scientific) was added to each sample and the reactions were incubated at 37°C for 20 min. Next, a subsample corresponding to 20% of each one of the cDNA libraries was used in a 50 µL test PCR reaction to determine the optimum number of cycles for library amplification using the RP1 RNA PCR primer (5'-AATGATACGGCGACCACCGAGATCTACACGTTCAGAGTTCTACAGTCCGA-3'), an RPIX RNA PCR primer primary index primer (5'-CAAGCAGAAGACGGCATACGAGATNNNNNNGTGACTGGAGTTCCTTGGCACCCGAGAATTCCA-3'), and PML PCR mix from the Tru-Seq RNA small Sample Prep Kit. Samples were taken every two cycles starting at cycle 16 and the PCR reactions were resolved on 2.5% agarose gels stained with SYBR Gold (Thermo Fisher Scientific). The optimum number of cycles was typically 22. After the determination of the optimum number of cycles, 50% of each cDNA library was amplified with indexing primers for deep sequencing. The amplified libraries were then purified using DNA Clean and Concentrator 5 columns (Zymo Research) and each one was eluted in 8 µL of elution buffer (Zymo Research). The amplified libraries were separated in 8% TBE gels against size standards for PAGE purification. The gels were stained with SYBR Gold and the gel portions containing the smears corresponding to the amplified libraries were excised (145–200 bp.). The gel slices were placed in nested tubes for gel crushing as described above. 0.6 mL of DNA elution buffer (10 mM Tris pH 8.0, 1 mM EDTA, 300 mM NaCl) were added to the crushed gel fragments and the slurry was incubated overnight at 25°C, shaking. The slurries were transferred to Spin-X 0.45 µm tube filters and centrifuged at 21,000 × g for 3 min to remove the gel fragments. The filtrates were transferred to fresh RNase-free non-stick microcentrifuge tubes and the DNA was precipitated with 1 volume of ice-cold isopropanol, and 15 µg Glycoblue for 60 min at -80°C. DNA pellets were recovered by centrifugation at 21,000 × g for 30 min at 4°C in a tabletop microcentrifuge. The pellets were washed with 0.75 mL 80% ice-cold ethanol, air-dried and re-suspended in 12 µL of 10 mM Tris pH 8.0. The quantity and quality of the cDNA libraries was assessed on a 2100 Bioanalyzer Instrument (Agilent Technologies) using a high-sensitivity DNA chip, and the libraries were diluted for 50-base-pair single-read deep sequencing on an Illumina HiSeq 2500 instrument. Approximately 110–170 million (AVG. $140 \times 10^6$) reads per library were recovered, covering ~5,500–8,500 megabases (~1.8–2.8× coverage of the human genome).

## RNA-Seq

All RNA-Seq experiments were performed as biological duplicates following the protocol below.

HEK293Trex cells were treated with doxycycline to induce expression of epitope-tagged IRE1, and subsequently treated with 5 µg/mL tunicamycin for 4 hr as per the modified PAR-CLIP protocol above. Total RNA was extracted from the cells using TRIzol (Thermo Fisher Scientific) following the manufacturer's recommendations. For each sample, ribosomal RNA was removed from 5 µg of DNase I (New England Biolabs)-treated total RNA using the Ribo-Zero Gold rRNA removal kit (Illumina) following the manufacturer's recommendations. The rRNA-depleted samples were then used to prepare directional libraries for deep sequencing using the ScriptSeq RNA-Seq v2 Library Preparation Kit (Illumina) following the manufacturer's recommendations, except the libraries were PAGE purified using 8% TBE gels instead of using solid phase reversible immobilization (SPRI) beads. The

quantity and quality of the libraries was assessed on a 2100 Bioanalyzer Instrument (Agilent Technologies) using high-sensitivity DNA chips, and the libraries were diluted for 50-base-pair single-read deep sequencing on an Illumina HiSeq 2500 instrument. Approximately 50–60 million reads per library were recovered, covering ~2,500–3,000 megabases (~0.8–1.0× coverage of the human genome).

## Computational analyses of next generation sequencing data
### PAR-CLIP read alignment, cluster annotation, and metrics

Deep sequencing reads from PAR-CLIP experiments were stripped of the RA3 cloning adapter sequences using in-house scripts. The adapter-stripped reads were uploaded to the CLIPZ analysis environment (*Khorshid et al., 2011*) at http://www.clipz.unibas.ch for alignment and analysis. Reads shorter than 15 nt long were discarded, and the remaining reads were aligned to the UCSC hg19 version of the human genome. After the first-pass alignment, the annotated reads were retrieved from the CLIPZ servers, and all the reads that were annotated as 'bacterial', 'fungus', 'vector', 'none', as well as all the unmapped reads, were removed from the original sequencing data using in-house scripts. The resulting sequence files devoid of blacklisted and unmapped sequences, containing between 3 and 5.7 million reads (~3%, on average, of the original reads), were then re-uploaded to CLIPZ for re-alignment to the reference genome and analysis. The breakdown of RNA classes recovered after mapping, mutational spectrum analyses, and mRNA regional preference analysis were obtained with the validated CLIPZ tools. Next, the mapped read clusters were retrieved from the CLIPZ servers for further metrics and analysis. For all transcripts (mRNAs, ncRNAS, miRNAs and small cytosolic RNAs; excluding tRNAs, rRNAs, and *RN7SL* pseudogenes), the CLIPZ-generated data was used to create lists of top hits, with the respective quantification of the numbers of crosslinked copies in each condition, as follows: First, the numbers of reads for every cluster containing at least three reads, where at least one of the reads possess a T→C mutation, were summed up for every annotated transcript locus, in every experimental condition, in each biological replicate, yielding a '*copy number*' per transcript. This method ensured that only those clusters that contain the stereotypical T→C mutant reads were included for subsequent analyses. Next, the geometric mean of the summed reads that satisfied the condition above was computed for each locus in each experimental condition (*i.e.*, IRE1 forced expression alone or chemically-induced ER stress on top of forced expression), thus generating summary tables in which we tabulated the copy number per transcript in biological duplicates for each experimental condition. After generating these summary tables, only those transcripts for which the copy number was at least five were considered for subsequent analysis. This method ensures that transcripts that have a single cluster with mutant reads, but in which the cluster contains less than five reads are not considered hits. By the same rationale, to be considered hits, transcripts holding mutant-read containing clusters made up of less than five reads, must have multiple clusters mapped onto them, so that the sum of the reads over the body of the transcript –*copy number*– is equal to or exceeds 5. Last, read clusters composed of only T→C mutant reads were also discarded, thus ensuring that mapped reads arising from naturally occurring single nucleotide polymorphisms, or from the spurious introduction of mutations during library preparation, were not considered to have been originated from crosslinked RNAs. After applying all these filters, the remaining transcripts were considered PAR-CLIP hits for all subsequent analyses, and their enrichment was estimated from the number of crosslinked copies –*copy number*– as defined per the rules above. Read cluster files containing reads mapping to tRNAs were cleaned-up and re-clustered using in-house scripts to allow annotation of wild-type and mutant read copies on known human tRNAs in the GtRNAdb database at http://gtrnadb.ucsc.edu. Next, all crosslinked tRNAs were grouped by codon-anticodon and the numbers of crosslinked copies, defined as per the rules above, were summed up for each codon-anticodon group. Codon usage for *Homo sapiens* was obtained from the codon usage database at http://www.kazusa.or.jp/codon/. Because of their abundance, read clusters for rRNA were not processed as described above, and the crosslink sites were extracted both by using the CLIPZ tool for genome site extraction and by visual inspection of read coverage tracks. Orthogonal validation of the crosslink sites on hit transcripts was accomplished by visual inspection of read coverage tracks mapped with a second aligner. For this purpose, the sequencing libraries were re-aligned to the hg19 version of the human genome using the sequence aligner Bowtie2 V2.1.0 (*Langmead et al., 2009*). For these analyses, the reads were stripped of the

RA3 cloning adapter sequences using in-house scripts as described above, and reads shorter than 13 ntlong were discarded. The remaining reads were aligned to hg19 Bowtie indices using the following options: -D 15 R 2 N 1 L 15 -i S,1,1.5 –end-to-end, and Sequence Alignment Map (SAM) format files were generated. The Bowtie2-generated SAM files were then converted to their binary (BAM) equivalent, sorted and indexed using SAMTools (*Li et al., 2009*). The resulting sorted BAM files were uploaded onto the Broad Institute Integrative Genomics Viewer (IGV) V2.3.67 (*Robinson et al., 2011*), for visualization. These alignments yielded similar results to those obtained with CLIPZ (~3%, on average, of the reads mapped once to the genome).

## RNA-Seq read alignment, annotation, and metrics

Deep sequencing reads from RNA-Seq experiments were stripped of the 3' cloning adapter sequences (5'-AGATCGGAAGAGCACACGTCTGAAC-3') using in-house scripts. Reads shorter than 18 nt long were discarded. The adapter-stripped reads were then aligned to hg19 Bowtie indices using the splice junction mapper TopHat2 V2.0.13 (*Kim et al., 2013*) and the sequence aligner Bowtie2 V2.2.3.0 (*Langmead and Salzberg, 2012*), using default parameters and specifying the library type to –library-type=fr secondstrand. The accepted lists of mapped reads in BAM format (per condition, per experiment) were then used as an input to assemble and quantify transcripts (on a per condition, per experiment basis) with the transcript assembler Cufflinks V2.1.1 (*Trapnell et al., 2010*), using an hg19 reference annotation in Gene Transfer Format (GTF, option -G), a list of sequences to be ignored composed of rRNA, tRNA and mitochondrial sequences (in GTF format, -M option), an hg19 sequence file in FASTA format for bias correction (-b option), and the following options: -u -m 50 s 2 –upper-quartile-norm –compatible-hits-norm –library-type=fr secondstrand. To ascertain the identities of all transcripts present across experiments, the resulting assembled transcriptomes generated by Cufflinks (in GTF format) were compared between biological replicates using Cuffcompare (*Trapnell et al., 2012*) with the -R option and an hg19 reference annotation in GTF format (-r option). Finally, to estimate the changes in gene expression levels, the Cufflinks-quantified transcripts were analyzed with Cuffdiff (*Trapnell et al., 2013*) using a list of sequences to be ignored composed of rRNA, tRNA and mitochondrial sequences (in GTF format, -M option), an hg19 sequence file in FASTA format for bias correction (-b option), and the following options: –FDR=0.1 compatible-hits-norm –upper-quartile-norm –library-type=fr secondstrand –dispersion-method=per condition -m 50 s 2 c 5. This analysis yielded transcript level estimates measured as the average fragments per kilobase per million (FPKM) per experimental condition which were used for gene expression profiling of each one of the transcripts and samples utilized in this study.

## Other computational analyses

### Heat maps

For visualization of the relative amounts of crosslinked transcripts in each condition, transcripts whose average copy number was less than 10 as per the rules above were excluded from the analysis. Then, the copy numbers of each transcript in every condition (*i.e.*, IRE1 forced expression alone or chemically-induced ER stress on top of forced expression) were divided by the median transcript copy number obtained in control conditions (IRE1 forced expression alone). These median-normalized numbers were log base-2-transformed and utilized to cluster the genes using Cluster V3.0 (*de Hoon et al., 2004*). The clusters were then assembled onto heat maps for visualization using Java TreeView V1.1.6r3 (*Saldanha, 2004*). Heat maps for visualization of the relative amounts of tRNAs (grouped by codon-anticodon), were generated in the same way.

### Grouping of genes traversing the secretory pathway

To ascertain the identities of those genes that traverse the secretory pathway, all the protein-coding genes annotated in the HUGO Gene Nomenclature Committee at http://www.genenames.org were cross-referenced to lists of genes whose transcripts encode signal peptides, transmembrane passes, or secreted proteins, obtained from both the Universal Protein Resource (UniProt) knowledge database at http://www.uniprot.org/uniprot/ and the human secretome and membrane proteome annotated in the Human Proteome Atlas at http://www.proteinatlas.org/humanproteome/secretome. The lists of PAR-CLIP hits were then cross-referenced to the list above and the proportion of hits traversing the secretory pathway was computed.

## Gene functional category enrichment analyses

Only PAR-CLIP protein-coding hit genes were used for gene functional category enrichment analyses. Enrichment P-values were calculated using the default settings of the DAVID (*Huang et al., 2009*). Functional Annotation Clustering tool at https://david.ncifcrf.gov. All the annotated protein coding genes in the *H. sapiens* genome were used as background. The analyses were focused on UniProt sequence features and Gene Ontology (GO) terms, as defined within the functional categories in the clustering tool. The generated lists were then curated manually. To eliminate redundant functional categories, only those categories with the more significant P-value, describing a larger set of genes, were taken into consideration.

## Correlations and estimation of statistical significance

Pearson correlation coefficients between (i) PAR-CLIP biological replicates, (ii) PAR-CLIP hits and gene length, and (iii) PAR-CLIP and RNA-Seq hits, and the estimation of the respective statistical significance, were computed using GraphPad Prism V6.0 (GraphPad Software Inc.).

## Immunoprecipitation and proteomics

### Native immunoprecipitation

20 million HEK293Trex cells were lysed with 250 μL of Lysis Buffer: 25 mM HEPES pH 7.4 150 mM NaCl, 1% NP-40 (or 1.4% digitonin), 1 mM EDTA, 10% Glycerol, phosphatase inhibitor (Roche) and 2× protease inhibitor cocktail (Roche) both freshly added. The cells were lysed by vortexing 3 sec with 3 min interval for 10 min. Cell debris was removed through centrifugation at 13,000 rpm for 15 min. For every 20 million cells, 5 μg of anti-Flag M2 antibody (Sigma) or anti-IRE1 antibody were used. The antibodies were, coupled to 40 μL of Protein G Dynabeads for 20 min at room temperature. After washing out the uncoupled antibody three times with lysis buffer, the lysate was incubated with antibody coupled Dynabeads for 4 hr at 4°C. After binding, the samples were washed 5 times with 500 μL lysis buffer and eluted by either boiling in 50 μL of 1× sample buffer for 5 min or incubated with 100 μL of 100 μg/mL 3× Flag peptide for 25 min at room temperature. For mass spectrometry analyses, the samples were washed 5 additional times with 500 μL of lysis buffer without detergent to remove the residual detergent followed by trypsin digestion on the Dynabeads. For native mass spectrometry experiments, around 200 million cells per condition were used, in triplicates.

### Mass spectrometry

Proteins were reduced with 1 mM DTT for 30 min, alkylated with 5.5 mM iodoacetamide for 20 min in the dark, and digested for 3 hr at room temperature with the endoproteinase LysC. Samples were diluted four times with ABC buffer (50 mM ammonium bicarbonate in $H_2O$, pH 8.0) and digested with trypsin overnight at 37°C. Acidified peptides were desalted by StageTip purification (*Rappsilber et al., 2007*). Samples were eluted with 60 μL of buffer B (80% ACN, 0.1% formic acid in $H_2O$) and reduced in a Vacufuge plus (Eppendorf) to a final volume of 3 μL. Buffer A (2 μl) (0.1% formic acid in $H_2O$) was added, and the resulting 5 μL were injected for reversed-phase chromatography on a Thermo Easy nLC 1000 system connected to a Q-Exactive mass spectrometer (Thermo) Peptides were separated on 15 cm columns (New Objective, Woburn, MA) with an inner diameter of 75 μm packed in house with 1.9 μm C18 resin (Dr. Maisch GmbH, Ammerbuch-Entringen, Baden-Würtemberg, Germany). Peptides were eluted with a linear gradient of acetonitrile from 5–27% in 0.1% formic acid for 95 min at a constant flow rate of 250 nL/min. The column temperature was kept at 35°C in an oven (Sonation GmbH, Biberach, Baden-Württemberg, Germany) with a Peltier element. Eluted peptides from the column were directly electrosprayed into the Q-exactive mass spectrometer via a nanoelectrospray source (Thermo). Mass spectra were acquired on the Q-Exactive in data-dependent mode automatically switching between full scan MS and up to ten data-dependent MS/MS scans. The maximum injection time for full scans was 20 msec, with a target value of 3,000,000 at a resolution of 70,000 at m/z = 200. The ten most intense multiply charged ions ($z \geq 2$) from the survey scan were selected with an isolation width of 1.6 Th and fragmented with higher energy collision dissociation (*Olsen et al., 2007*) with normalized collision energies of 25. Target values for MS/MS were set at 1,000,000 with a maximum injection time of 60 msec at a resolution of

17,500 at m/z = 200. To avoid repetitive sequencing, the dynamic exclusion of sequenced peptides was set at 20 sec.

The resulting MS and MS/MS spectra were analyzed using MaxQuant (version 1.3.0.5), utilizing its integrated ANDROMEDA search algorithms (*Cox and Mann, 2008*; *Cox et al., 2011*). Peak lists were searched against local databases for human proteins concatenated with reversed copies of all sequences. The search included carbamidomethlyation of cysteine as a fixed modification and methionine oxidation and N-terminal acetylation as variable modifications. The maximum allowed mass deviation was 6 ppm for MS peaks and 20 ppm for MS/MS peaks and the maximum number of missed cleavages was 2. The maximum false discovery rate was 0.01 on both the peptide and the protein level and was determined by searching a reverse database. The minimum required peptide length was 6 residues. Proteins with at least 2 peptides (one of them unique) were considered hits. The 'match between runs' option was enabled with a time window of 2 min to match identification between samples.

In UV crosslinking experiments proteins not identified in a sample were assigned an arbitrary low intensity of 5 to allow ratio calculations. Intensity ratios were calculated for thapsigargin-treated versus DMSO-treated cells that where UV crosslinked. Proteins reporting a ratio of higher than 10 and that reported at least 3 identified peptides were considered potential UPR induced interactors. Both cut-offs were set arbitrarily.

Label-free quantitation was done with the QUBIC software package as described elsewhere (*Hubner et al., 2010*). All calculations and plots were done with the R software package (http://r-project.org/).

## qRT-PCR

HEK293T or RPMI-8226 cells were cultured and treated as described above or in figure legends. All experiments were performed on 6-well tissue culture plates. Sub-confluent cells from each well were collected in 1 mL of TRIzol reagent, and total RNA was extracted. 500 ng of total RNA were reverse transcribed using the SuperScript VILO system (Thermo Fisher Scientific) following manufacturer's recommendations. The resulting 20 μL reactions containing cDNAs were diluted to 200 μL with 10 mM Tris pH 8.2 and 2 μL of this dilution were used as template for each quantitative real time PCR using IQ SYBR Geen Super Mix (BioRad) in 20 μL reactions. The reactions were ran on a BioRad CFX96 Real Time system (BioRad) and analyzed using the CFX Manager Software V3.0 (BioRad). All reactions were normalized to an internal loading control (GAPDH). The following oligonucleotides targeting human transcripts were used:

| Target | Fwd. primer | Rev. primer |
| --- | --- | --- |
| *Hs BLOC1S1* | 5'-AGCTGGACCATGAGGTGAAG-3' | 5-AGCTGGACCATGAGGTGAAG-3' |
| *Hs CALR* | 5'-CCACCCAGAAATTGACAACC-3' | 5'-TTAAGCCTCTGCTCCTCGTC-3' |
| *Hs GAPDH* | 5'-AGCCACATCGCTCAGACAC-3' | 5'-TGGAAGATGGTGATGGGATT-3' |
| *Hs HSPA5* | 5'-TGCAGCAGGACATCAAGTTC-3' | 5'-AGTTCCAGCGTCTTTGGTTG-3' |
| *Hs RN7SL 5'Alu* | 5'-CGCTTGAGTCCAGGAGTTCT-3' | 5'-GTTTTGACCTGCTCCGTTTC-3' |
| *Hs RN7SL S domain* | 5'-ATCGGGTGTCCGCACTAA-3' | 5'-ACTGATCAGCACGGGAGTTT-3' |
| *HS RN7SL 3' Alu* | 5'-ATCGGGTGTCCGCACTAA-3' | 5'-TGGAGTGCAGTGGCTATTCA-3' |
| *Hs SSR3* | 5'-TGGAAGAAGAATGAAGTTGCTG-3' | 5'-CTTGTGTCTCCCACCCTGAC-3' |
| *Hs TMED7* | 5'-GCCTCCAAAAATGGGACATA-3' | 5'-GCCCTATGCTAACCACCAGA-3' |
| *Hs TMEM109* | 5'-GCCTTCTTTGCTCTGTCTGG-3' | 5'-GATCAGCAAGGCCAGGAGT-3' |
| *Hs spliced* | 5'-AGCTTTTACGAGAGAAAACTCAT-3' | 5'-CCTGCACCTGCTGCG-3' |

## Western blots

Cell lysates were fractionated by sequential detergent extraction as outlined in the modified PAR-CLIP protocol, or immunoprecipitates were collected in native conditions as described above. Samples were then mixed with 2× SDS PAGE sample buffer, heated to 95°C for 5 min and separated on Tris-glycine PAGE gels. 2-mercaptoethanol (2ME) was added to a final concentration of 5% to the lysates just prior to boiling and loading on SDS-PAGE gels. The proteins were then transferred to

0.2 μm pore size nitrocellulose membranes and blocked with 5% non-fat dry milk, 0.1% Tween 20 in Tris-buffered saline. The blocked membranes were then probed with the following antibodies (diluted in 5% BSA in Tris-buffered saline, 0.1% Tween 20): anti-FLAG mouse monoclonal antibody (M2, Sigma-Aldrich, 1:1,000), anti-GAPDH rabbit polyclonal antibody (Abcam ab9485, 1:2,000), anti-IRE1 rabbit monoclonal antibody (Cell Signaling Technology 14C12, 1:1,000), anti-NONO mouse monoclonal antibody (Santa Cruz Biotechnology p54/nrb A-9, sc-166704, 1:1,000), anti-RPL13A antibody (Cell Signaling Technology, 1:1,000), anti-SRP19 antibody (Proteintech, 16033-AP-1, 1:1,000), anti SRP14 antibody (Proteintech, 11528-AP-1, 1:1,000), anti-SRP54 antibody (BD Transduction Laboratories, 1:1,000), anti-SEC61 antibody (Cell Signaling Technology, 1:1,000), anti-RPS15 antibody (Proteintech, 14957–1-AP, 1:1,000). Immunoreactive bands were detected using HRP-conjugated secondary antibodies (Amersham, GE Healthcare Life Sciences NA931, NA934, 1:5,000) and luminol-based enhanced chemiluminescence substrates (SuperSignal West Dura Extended Duration Substrate, Life Technologies) and exposed to radiographic film or imaged directly in a digital gel imager (LI-COR Odyssey, LI-COR Biosciences or Chemidoc XRS+, BioRad). Digital images were automatically adjusted for contrast using the photo editor Adobe Photoshop (Adobe Systems).

## Northern blots

HEK293T or RPMI-8226 cells were cultured and treated as described above or in figure legends and total RNA was extracted with TRIzol reagent as described above. 1 μg of total RNA was mixed with excess 2× Novex TBE-urea sample buffer (Thermo Fisher Scientific), heated at 95°C for 5 min, then placed immediately on ice and loaded onto either 6% or 15% TBE-urea PAGE gels (Thermo Fisher Scientific) to blot for SRP RNA or tRNAs respectively. The gels were stained with SYBR Gold and imaged in a digital gel imaging system (Chemidoc XRS+, BioRad) to check for equal loading. The RNA was then transferred onto positively charged Hybond-N+ nylon membranes (GE Healthcare) by electroblotting in pre-chilled 0.5× TBE at a constant voltage of 80V for 60 min using a wet blot transfer apparatus (BioRad). To immobilize the RNA, the membranes were photocrosslinked using 150 mJ/cm$^2$254 nm UV light in a GS Gene Linker UV chamber (BioRad). The crosslinked membranes were then imaged in a Chemidoc XRS+ digital imager to check for even transfer. Next, the membranes were pre-hybridized in hybridization buffer (0.5 M $Na_2HPO_4$ pH 7.2, 1 mM EDTA, 7% SDS and 10 g/L BSA) at 65°C (for SRP RNA) or 42°C (for tRNAs) for 2 hr in a rotating incubator. Then, radiolabelled probes were added and the membranes were hybridized overnight at 65°C or 42°C. The hybridized membranes were washed with pre-warmed wash buffer (2× SSC, 0.1% SDS; at the respective hybridization temperatures) on a rotating platform until no residual radioactivity was detected in the wash buffer using a Geiger-Müller counter equipped with a pancake probe (2–3 washes). The radiolabeled probes for Northern blots were prepared as follows: (i) tRNA probes: 30 pmol of 5'-ends DNA oligonucleotides (antisense to the target tRNAs) were end-labeled with 50 μCi of γ-$^{32}$P-ATP (3,000 Ci/mmol, 10 mCi/ml) and 25 units of T4 PNK (New England Biolabs) in 50 μL reactions for 30 min. at 37°C. (ii) 50 ng of *in vitro* annealed long DNA oligonucleotides encoding a portion of the *RN7SL* S-domain were used to prepare body-labeled DNA probes using 50 ng/μL random hexamers (Thermo Fisher Scientific), 1 mM dATP, 1 mM dGTP, 1 mM dTTP, 50 μCi of α-$^{32}$P-dCTP (3,000 Ci/mmol, 10 mCi/ml; Perkin Elmer), 5 units of Klenow fragment (3' → 5' exo-) (New England Biolabs) in 50 μL reactions set-up in NEB Buffer 2 (10 mM Tris pH7.9, 50 mM NaCl, 10 mM $MgCl_2$ and 1 mM DTT; New England Biolabs). In both cases, the reactions were terminated by addition of EDTA to a final concentration of 10 mM, followed by heating at 95°C for 3 min and placing the reactions on ice immediately afterwards. The radiolabeled probes were cleaned-up of unincorporated label using Illustra MicroSpin G25 size-exclusion columns (GE Healthcare) following the manufacturer's recommendations. The following oligonucleotides were used to generate probes for Northern blots:

| Target | Oligonucleotide (5' to 3') |
| --- | --- |
| tRNA Gln 3' end | GGTCCCACCGAGATTTGAACTCGG |
| tRNA-CUG (CAG anticodon) | AGGTTCCACCGAGATTTGAACTCGGATCGCTGGATTCA GAGTCCAGAGTGCTAACCATTACACCATGGAACC |

*Continued on next page*

*Continued*

| Target | Oligonucleotide (5' to 3') |
|---|---|
| RN7SL S-domain (sense) | GATCGGGTGTCCGCACTAAGTTCGGCATCAATATGGTG ACCTCCCGGGAGCGGGGGACCACCAGGTTGCCTAAGGAGGGG |
| RN7SL S-domain (antisense) | CCCCTCCTTAGGCAACCTGGTGGTCCCCCGCTCCCGGGAG GTCACCATATTGATGCCGAACTTAGTGCGGACACCCGATC |

## Purification of IRE1-KR

The IRE1 cytosolic domain was purified according to published protocols (*Li et al., 2010*). Briefly, human IRE1-KR43 bearing an N-terminal 6× His tag was expressed in SF21 insect cells using baculovirus system. The pellet from 250 mL of SF21 cells was suspended in a 20 mL lysis buffer containing 20 mM HEPES pH 7.4, 600 mM KCl, 2 mM $MgCl_2$, 10% Glycerol, 10 mM imidazole, and EDTA-free protease inhibitor cocktail (Roche). The cells were broken by passing them through an Emulsiflex homogenizer set at 16,000 psi, once. The lysate was centrifuged at 16,000 rpm for 40 min in a SS-34 rotor to remove cell debris. The clarified cell lysate was loaded onto a 5 mL HisTRAP HP column (GE Healthcare) at a 2 mL/min flow rate. The lysate was then washed with 20 column volumes of lysis buffer and eluted with an imidazole gradient to 500 mM imidazole at 15 column volumes. The IRE1-KR43 eluate was then incubated with TEV protease to remove the 6× His tag overnight during dialysis against 20 mM HEPES pH 7.4, 300 mM KCl, 2 mM $MgCl_2$, 5% Glycerol, 1 mM TCEP. After TEV cleavage, the protein was loaded onto a HisTRAP HP column to remove the impurities and IRE1-KR43 with unremoved 6× His tag. The flow through of the HisTRAP HP column was further purified on Superdex 200 column (GE Healthcare), which was equilibrated with the dialysis buffer. The IRE1-KR43 eluted from Superdex 200 was then concentrated to 50 µM and frozen in small aliquots.

## Purification of 80S ribosomes and ribosomal subunits

For splitting ribosomal subunits, a 5 mL pellet of K562 cells was suspended in 10 mL of Lysis Buffer (20 mM HEPES, pH 7.4, 100 mM KCl, 5 mM $MgCl_2$, 1 mM TCEP, 0.5% NP-40 and complete EDTA-free protease inhibitor cocktail (Roche), 0.5 units/µl RNasin (Promega) and lysed by short intervals of vortexing (10 min). Cell debris was removed by centrifugation for 15 min at 15,000 × *g* at 4°C. To split the ribosomes, the salt concentration was adjusted to 500 mM KCl and the lysate was incubated with 1 mM puromycin for 30 min on ice, then for 20 min at 20°C (adapted from Blobel and Sabbatini). The ribosomes were pelleted through a high-salt sucrose cushion: 1M sucrose, 500 mM KCl, 5 mM $MgCl_2$, 1 mM TCEP, 0.5 mM PMSF at 33,000 rpm for 13.5 hr using a Ti70 rotor, via layering 12.5 mL lysate on 5 mL of cushion. The transparent ribosomal pellet was suspended in buffer 20 mM HEPES pH 7.4, 500 mM KCl, 5 mM $MgCl_2$, 1 mM TCEP, 0.5 mM PMSF, 0.5 units/µL RNasin and protease inhibitors. 200–250 µL of ribosomes were loaded onto a 10–40% sucrose gradient (10–40% sucrose in buffer: 20 mM HEPES, pH 7.4, 500 mM KCl, 5 mM $MgCl_2$, 0.5 mM PMSF) and centrifuged at 25,000 rpm for 12 hr using a SW40 rotor. Ribosomal subunits were pelleted from suitable fractions by centrifugation at 36,000 rpm for 14.5 hr using a TLA110 rotor (corresponds to 115,800 × *g*, modified from Khatter et al.,). The pellet was suspended in 20 mM HEPES, pH7.4, 100 mM KCl, 5 mM $MgCl_2$, 1 mM TCEP, 0.5 mM PMSF and protease inhibitor cocktail (Roche). The isolated 40S and 60S ribosomal subunits were either used in the depicted assays or reconstituted to 80S by incubation in low salt resuspension buffer (150 mM KCl). The reconstituted 80S was then run through a 10–40% sucrose gradient.

## Microscale thermophoresis experiments (MST)

MST experiments were performed with a Monolith NT.115 instrument (NanoTemper Technologies, Germany). For thermophoresis experiments IRE1-KR43 was labeled using Monolith Protein Labeling Kit RED-maleimide kit (cysteine reactive, NanoTemper) using the manufacturer's protocols. For the assays, 100 nM IRE1-KR43 labeled with RED dye were incubated with 80S at various concentrations for 30 min at 25°C. The measurements were done using Premium Capillaries (NanoTemper Technologies, Germany) at 20% LED power and 40% IR-laser at 25°C.

## Co-sedimentation assays

All the experiments were performed in Ribosome Binding Buffer (5 mM Hepes pH 7.4, 100 mM KCl, 5 mM MgCl$_2$, 1 mM TCEP). 100 nM IRE1-KR and ribosomes at various concentrations (25–500 nM) were incubated for 30 min on ice. 45 µL of each reaction were layered over 100 µL of sucrose cushion made with 1.25 M sucrose in Ribosome Binding Buffer. The ribosomes and their complexes were spun at 75,000 rpm for 60 min in a TLA100 rotor. The pellet was washed twice with 200 µL of ribosome binding buffer and resuspended in 25 µL of 1× SDS-buffer.

## Size exclusion chromatography

For size exclusion analyses, 50 µL of IRE1-KR (100 nM) and IRE1-KR-ribosome complexes at 1:1, 1:2 molar ratio were incubated on ice for 30 min before loading onto a Sephacryl 300 column (self-packed tricorn 5/100, GE Healthcare) equilibrated with 20 mM HEPES 7.4, 150 mM KCl, 5 mM MgCl$_2$ and 1 mM TCEP. 100 µL fractions were collected and loaded onto SDS-PAGE after trichloro-acetic acid precipitation.

## *In vitro* transcription of PAR-CLIP targets

The templates for *in vitro* transcription reactions were either amplified by PCR (SEC61A1, TMEM109, TMED7, DDOST) or by linearizing plasmids (ATPV0B, BLOC1S1). After gel purification, the linearized plasmid templates were cleaned up and concentrated to 500 ng/µL with Zymo Clean-UP and concentrator 5 columns. RNA was transcribed using HiScribe high yield T7 kit (New England Biolabs) for 2 hr at 37°C. The reactions were stopped by addition of DNaseI and incubation for 15 min at 37°C. The RNA products were PAGE separated using 6% urea-PAGE gels (Invitrogen). The RNAs were excised from the gels with a razor blade and the gel fragments crushed into small pieces. The RNAs were extracted from the gels using 3 gel volumes of RNA extraction buffer (300 mM NaOAc, 1 mM EDTA, and RNasin prepared in RNAse-free water). For extraction, the gel slurry was incubated at 20°C shaking for an hour. The slurry was transfered to a Spin-X 0.45 µm tube filter and spun at 15,000 rpm for 3 min to filter out the gel pieces. The RNAs were precipitated from the filtrate with isopropanol, air dried and resuspended in RNase-free water. The transcribed RNAs were then folded by incubating them for 5 min at 95°C and cooling down to 25°C at 1 °C/min in a thermocycler.

| Target | Fwd. primer | Rev. primer |
|---|---|---|
| *Hs ATP6V0B* | 5'-TAATACGACTCACTATAGGGCATCGGAACTACCATGCAGGC-3' | 5'-GCACTGACCCAGACAAATACC-3' |
| *Hs BLOC1S1* | 5'-AGCTGGACCATGAGGTGAAG-3' | 5-AGCTGGACCATGAGGTGAAG-3' |
| *Hs DDOST* | 5'-TAATACGACTCACTATAGGGCTCCCCTTCGGTAGAAGATT-3' | 5'-ATACTCCACTAGGTCAGTGACAGTG-3' |
| *Hs SEC61A1* | 5'-TAATACGACTCACTATAGGGGGAGCTAGGGATCTCTCCTAT-3' | 5'-CATTTGGGAGATGACATAAAGGTTGG-3' |
| *Hs TMED7* | 5'-ATGCCGCGGCCGGGGTCCGC-3' | 5'-TTATGATCCAACACGAGTTGTG-3' |
| *Hs TMEM109* | 5'-ATGGCAGCCTCCAGCATCAGTTC-3' | 5'-TCACTCCTCCTCCACACTGCG-3' |

| Gene | Plasmid ID | Construct |
|---|---|---|
| HsATP6V0B | pPW3005 | pDAA-pUC19-ATP6V0B(FL)—1 |
| HsBLOC1S1 | pPW3004 | pDAA-pUC19-BLOC1S1(FL)—1 |
| HsDDOST | pPW3061 | pMGC3161588 |
| HsSEC61A1 | pPW3316 | pSEC61A1-pcDNA3.1-flag(FL) |
| HsTMED7 | pPW3057 | pMGC3625342 |
| HsTMEM109 | pPW3062 | pDAA-TMEM-DT1 |

### *In vitro* cleavage assays with purified IRE1-KR

For *in vitro* cleavage assays, 50 nM to 5 µM IRE1-KR were incubated with the RNA of interest at 30°C for the indicated times, ranging from 30 sec to 60 min. The assays were performed in RNA cleavage buffer (25 mM HEPES pH 7.4, 150 mM KCl, 5 mM $MgCl_2$, 1 mM TCEP and 5% glycerol). The reactions were stopped by incubating with Proteinase K for 10 min at 37°C and then heating them after addition of stop buffer at 75°C for 5 min. IRE1's RNAse activity was inhibited by incubation with 10 µM 4µ8C for 20 min on ice. For the experiments conducted with IRE1-ribosome complexes, the RNAs were incubated with ribosomes alone to take into account unspecific RNAse activity carried over with the purified ribosomes, which was close to nonexistent. The samples were run on 6% TBE-urea PAGE gels for 1.5 hr at 100 V.

### Biolayer interferometry experiments

The biolayer Interferometry measurments were conducted using OCTET RED384 system (Fortebio). All the experiments were conducted at 25°C in using the following the buffer: 25 mM HEPES pH 7.4, 150 mM KCl, 5 mM $MgCl_2$, and 250 µM TCEP. IRE1 KR with N-terminal 6× His-tag is at 100 nM and 1 µM concentrations was coupled to Ni-NTA biosensors for 60–120 secs (ForteBio). The data obtained from empty sensors were subtracted from IRE1-KR coupled sensors to eliminate background binding. 80S and 60S ribosomal subunits at concentrations ranging from 5 to 500 nM were used to monitor their binding to and dissociation from IRE1-KR. The data were fitted using OCTET data analysis software version 10.

### Peptide arrays

Peptide arrays were purchased from the MIT Biopolymers Laboratory. The tiling arrays were composed of 18-mer peptides that were tiled along the MGAT2, SSR3, TMEM109, TMED7 protein sequences with a five amino acid shift between adjacent spots. The arrays were incubated in 100% methanol for 10 min, then in Binding Buffer (50 mM HEPES pH 7.2, 150 mM NaCl, 0.02% Tween-20, and 2 mM DTT) three times for 10 min each. The arrays were then incubated for 1 hr at room temperature with 500 nM MBP-hIRE1α cLD and washed three times with 10 min incubation in between the washes in the Binding Buffer to remove any unbound protein. The bound protein was transferred to a nitrocellulose membrane using a semi-dry transfer apparatus and detected with anti-MBP antiserum (New England Biolabs). The contribution of each amino acid to hIRE1α cLD was calculated as described previously (*Gardner and Walter, 2011*).

## Acknowledgements

We thank Dr. Alexander Mercier and Dr. Martin Kampmann for their help with bioinformatics analyses. We are grateful to the MIT Biopolymers Institute for synthesizing the peptide arrays. DAA was supported by an Irvington Postdoctoral Fellowship of the Cancer Research Institute. PW is an Investigator of the Howard Hughes Medical Institute.

## Additional information

### Funding

| Funder | Grant reference number | Author |
| --- | --- | --- |
| Howard Hughes Medical Institute | Investigator | Tobias C Walther<br>Peter Walter |
| Cancer Research Institute | Postdoctoral fellowship | Diego Acosta-Alvear |

The funders had no role in study design, data collection and interpretation, or the decision to submit the work for publication.

### Author contributions

Diego Acosta-Alvear, G Elif Karagöz, Conceptualization, Data curation, Formal analysis, Validation, Investigation, Visualization, Methodology, Writing—original draft, Writing—review and editing;

Florian Fröhlich, Resources, Formal analysis, Methodology; Han Li, Resources; Tobias C Walther, Resources, Supervision, Funding acquisition, Methodology; Peter Walter, Conceptualization, Supervision, Funding acquisition, Investigation, Writing—original draft, Writing—review and editing

### Author ORCIDs
Diego Acosta-Alvear http://orcid.org/0000-0002-1139-8486
G Elif Karagöz https://orcid.org/0000-0002-3392-2250
Florian Fröhlich https://orcid.org/0000-0001-8307-2189
Peter Walter http://orcid.org/0000-0002-6849-708X

### Decision letter and Author response
Decision letter https://doi.org/10.7554/eLife.43036.024
Author response https://doi.org/10.7554/eLife.43036.025

## Additional files

### Supplementary files
• Supplementary file 1. Transcripts recovered in PAR-CLIP experiments performed by pulling-down on epitope-tagged IRE1 in the presence or absence of chemically induced ER-stress.
DOI: https://doi.org/10.7554/eLife.43036.019

• Supplementary file 2. tRNAs recovered in PAR-CLIP experiments performed by pulling-down on epitope-tagged IRE1 in the presence or absence of chemically induced ER-stress.
DOI: https://doi.org/10.7554/eLife.43036.020

• Supplementary file 3. IRE1 native IP-MS and PAR-CLIP-MS enriched peptides.
DOI: https://doi.org/10.7554/eLife.43036.021

• Transparent reporting form
DOI: https://doi.org/10.7554/eLife.43036.022

### Data availability
All data analysed during this study are included in the manuscript and supporting files.

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
