## [Decision Letter]

Thank you for submitting your article "The unfolded protein response and endoplasmic reticulum protein targeting machineries converge on the stress sensor IRE1" for consideration by *eLife*. Your article has been reviewed by two peer reviewers, and the evaluation has been overseen by a Reviewing Editor and James Manley as the Senior Editor. The following individual involved in the review of your submission has agreed to reveal his identity: Alexei V Korennykh (Reviewer #1). The other reviewer remains anonymous.

The reviewers have discussed the reviews with one another and the Reviewing Editor has drafted this decision to help you prepare a revised submission. For your benefit, the reviews are attached in their entirety.

Summary:

Acosta-Alvear et al. use PAR-CLIP methods to identify specific RNAs that bind to IRE1, such as suboptimal RIDD targets and structural rRNAs. The manuscript reports a physical interaction between 80S ribosomes and the transmembrane kinase-RNase, Ire1. Examples of ribosome-binding host endoribonucleases have been previously described for bacteria. However, little is known about eukaryotic RNases that interact with the ribosome. This study of Ire1 provides the first well-documented example of a eukaryotic endoribonuclease that functions by forming a mechanistic complex with the translation machine.

Ire1 is activated by unfolded proteins entering the ER and serves for negative feedback relieving the stress. The model that SRP RNA/Ribosome/Ire1 form a complex that helps location and degradation of stress-causing mRNAs is very appealing.

Essential revisions:

The two reviewers were very positive about your paper, and comment that the "paper is very impressive in the experimental approaches used and scientific advances it provides", and "this is an innovative and broadly interesting study that I can recommend for publication.

The major issue that needs to be discussed is the small amount of Ire1 in the nuclear fraction (Figure 1—figure supplement 1B). Reviewer #1 suggested in the ensuing discussion that because there is some nuclear GAPDH, perhaps the nuclear-appearing Ire1 could have been carried over in unlysed cells, a rather typical fractionation artifact. However, GAPDH can also be found in the nucleus:https://www.ncbi.nlm.nih.gov/pmc/articles/PMC3084531/

*Reviewer #1:*

The manuscript by Diego Alvear et al. reports a physical interaction between 80S ribosomes and the transmembrane kinase-RNase, Ire1. Examples of ribosome-binding host endoribonucleases have been previously described for bacteria. However, little is known about eukaryotic RNases that interact with the ribosome. This study of Ire1 provides the first well-documented example of a eukaryotic endoribonuclease, which functions by forming a mechanistic complex with the translation machine.

Ire1 is activated by unfolded proteins entering the ER and serves for negative feedback relieving the stress. The model that SRP RNA/Ribosome/Ire1 form a complex that helps location and degradation of stress-causing mRNAs is very logical.

The mechanism is documented by several methods, from PAR-clip RNA-seq to biochemical measurement of Kd between Ire1 and the 80S ribosomes. The combination of complementary approaches is a technical strength of this work. Overall, this is an innovative and broadly interesting study that I can recommend for publication.

1) My major concern was that the ribosome is a highly charged large macromolecule known to bind proteins non-specifically. Therefore, studies that report new ribosome-binding proteins face the challenge of proving physiologically relevance. Here the authors show that 40S and 60S subunits do not bind, which strongly argues that the 80S binding to Ire1 is specific. Moreover, the value of Kd of 30 nM suggests that under cellular ribosome concentrations of 1-10 uM, Ire1 will be with high probability bound to the ribosomes. Further supporting this argument Ire1 cross-links to a specific site on the 80S ribosome. Therefore, my key concern has been already addressed.

2) The authors do not see a drop in levels for any Ire1-crosslinked mRNAs (subsection “IRE1 association and RNA abundance do not correlate”, second paragraph). They explain this by the note that captured mRNAs have higher stability, leading to better cross-linked adduct survival and easier detection. The Ire1-bound mRNAs are therefore location markers rather than true clients.

The model proposed in this paper would become stronger if ribosome-dependent mRNA decay by Ire1 was functionally demonstrated. However, I understand that such a demonstration is not just a control, but a major standalone investigation and therefore I do not propose this work, unless the authors already have the data.

The central message that Ire1 forms a specific complex with the ribosome is not affected by this issue.

*Reviewer #2:*

Acosta-Alvear et al. use PAR-CLIP methods to identify specific RNAs that bind to IRE1, such as suboptimal RIDD targets and structural rRNAs. The paper is very impressive in the experimental approaches used and scientific advances it provides. I favour rapid publication of the manuscript in present form. I only have one issue:

- In Figure 1—figure supplement 1B, it is shown that a significative amount of Ire1 is present in the nuclear fraction. Maybe this is an artefact of the Ire1 overexpression system it is used. Maybe not. Other authors, as for example, Lee et al., 2002, Genes and Dev, Figure 5B, have reported Ire1 presence in LaminB receptor rich fraction. Presumably, such nuclear membrane localised Ire1 would be at the inner nuclear membrane, with the "cytosolic" domain of Ire1 in the nucleoplasm. It would be interesting, I think, to do PAR-CLIP from nuclear fraction IPed Ire1. In the worst case scenario, this experiment would be a nice negative control from the membrane fraction results. But what if, for example, Xbp1 mRNA is strongly PAR-CLIPed from the nuclear fraction (in contrast to failure in identifying Xbp1 mRNA in the membrane fraction PAR-CLIP results)?

This would be a very interesting result.

---

## [Author Response]

Reviewer #1:

[…] 1) My major concern was that the ribosome is a highly charged large macromolecule known to bind proteins non-specifically. Therefore, studies that report new ribosome-binding proteins face the challenge of proving physiologically relevance. Here the authors show that 40S and 60S subunits do not bind, which strongly argues that the 80S binding to Ire1 is specific. Moreover, the value of Kd of 30 nM suggests that under cellular ribosome concentrations of 1-10 uM, Ire1 will be with high probability bound to the ribosomes. Further supporting this argument Ire1 cross-links to a specific site on the 80S ribosome. Therefore, my key concern has been already addressed.

We appreciate the evaluation and are glad to know that our data have already addressed this particular concern.

2) The authors do not see a drop in levels for any Ire1-crosslinked mRNAs (subsection “IRE1 association and RNA abundance do not correlate”, second paragraph). They explain this by the note that captured mRNAs have higher stability, leading to better cross-linked adduct survival and easier detection. The Ire1-bound mRNAs are therefore location markers rather than true clients.The model proposed in this paper would become stronger if ribosome-dependent mRNA decay by Ire1 was functionally demonstrated. However, I understand that such a demonstration is not just a control, but a major standalone investigation and therefore I do not propose this work, unless the authors already have the data.The central message that Ire1 forms a specific complex with the ribosome is not affected by this issue.

We agree. Demonstrating ribosome-dependent IRE1 decay is beyond the scope of this work. A thorough investigation of such mechanism will require detailed follow-up studies, which have not yet been performed.

Reviewer #2:

[…] - In Figure 1—figure supplement 1B, it is shown that a significative amount of Ire1 is present in the nuclear fraction. Maybe this is an artefact of the Ire1 overexpression system it is used. Maybe not. Other authors, as for example, Lee et al., 2002, Genes and Dev, Figure 5B, have reported Ire1 presence in LaminB receptor rich fraction. Presumably, such nuclear membrane localised Ire1 would be at the inner nuclear membrane, with the "cytosolic" domain of Ire1 in the nucleoplasm. It would be interesting, I think, to do PAR-CLIP from nuclear fraction IPed Ire1. In the worst case scenario, this experiment would be a nice negative control from the membrane fraction results. But what if, for example, Xbp1 mRNA is strongly PAR-CLIPed from the nuclear fraction (in contrast to failure in identifying Xbp1 mRNA in the membrane fraction PAR-CLIP results)?This would be a very interesting result.

We agree that it is possible that a nuclear-membrane bound pool of signaling IRE1 may be targeted by different mRNAs, and perhaps even by XBP1. We agree that performing IRE1 PAR-CLIP on the nuclear fraction could indeed be an interesting experiment, and in the worst-case scenario, it would be a good negative control. However, since the PAR-CLIP experiments required a substantial amount of starting material, it would be a monumental undertaking to perform these experiments in cell factions that contain only a minor portion of the cell’s IRE1. The small amount of nuclear IRE1 observed in Figure 1—figure supplement 1B could represent carry-over from unlysed cells or a small pool of IRE1 resident in the nuclear envelope. The latter possibility remains plausible, because GAPDH can also be found in the nucleus (PMID:20727968). Given that our main discovery is the association of IRE1 with the ER co-translational protein targeting machinery, a detailed analysis of subcellular IRE1 pools would lie outside the scope of our study. We thank reviewer 2 for supporting the rapid publication of our work in present form.